# Dual adaptive differential threshold method for automated detection of faint and strong echo features in radar observations of winter storms

Laura M. Tomkins[1], Sandra E. Yuter[1,2], and Matthew A. Miller[2]

[1]Center for Geospatial Analytics, North Carolina State University, Raleigh, NC, 27695, USA
[2]Department of Marine, Earth and Atmospheric Science, North Carolina State University, Raleigh, NC, 27695, USA

**Correspondence:** Laura M. Tomkins (lmtomkin@ncsu.edu)

**Abstract.** Radar observations of winter storms often exhibit locally-enhanced linear features in reflectivity, sometimes labeled as snow bands. We have developed a new, objective method for detecting locally-enhanced echo features in radar data from winter storms. In comparison to convective cells in warm season precipitation, these features are usually less distinct from the background echo and often have more fuzzy or feathered edges. This technique identifies both prominent, strong features and more subtle, faint features. A key difference from previous radar reflectivity feature detection algorithms is the combined use of two adaptive differential thresholds, one that decreases with increasing background values and one that increases with increasing background values. The algorithm detects features within a snow rate field rather than reflectivity and incorporates an under and over estimate of feature areas to account for uncertainties in the detection. We demonstrate the technique on several examples from the US National Weather Service operational radar network. The feature detection algorithm is highly customizable and can be tuned for a variety of datasets and applications.

## 1 Introduction

Linear features of enhanced reflectivity, labeled as snow bands, are often observed in winter storms and are an active topic of research (Baxter and Schumacher, 2017; Ganetis et al., 2018; Lackmann and Thompson, 2019; Kenyon et al., 2020; Picca et al., 2014; Novak et al., 2004; McMurdie et al., 2022; Colle et al., 2023). Snow bands that are $\geq 250$ km in length are described as primary or single bands and sets of roughly parallel smaller bands each less than 250 km long are described as multi-bands (Ganetis et al., 2018). Primary bands are typically associated with frontogenesis (Novak et al., 2004), but the forcing mechanism for multi-bands is still unclear (Ganetis et al., 2018). Unlike convective cells in rain which usually have high reflectivities and a sharp reflectivity gradient between the cell itself and the background reflectivity, snow bands have weaker reflectivities, stand out less from the background, and the edges of snow bands can gradually taper out, creating an irregular edge. Hence, objective methods to identify convective and stratiform precipitation in radar data of deep convection do not work well for winter storms.

Much of the previous work to detect snow bands in radar reflectivity data focused on identification of primary bands and either ignored multi-bands or only addressed the stronger subset of multi-bands. Novak et al. (2004) and Baxter and Schumacher

(2017) used $\mathrm{dBZ}$ thresholds (30 and 25 $\mathrm{dBZ}$, respectively) to identify primary band objects in National Weather Service (NWS) Next Generation Radar (NEXRAD) Level-III reflectivity regional maps. Kenyon et al. (2020) identified primary snow bands for five winter seasons using Level-III reflectivity data. Kenyon et al. (2020) used a 20 $\mathrm{dBZ}$ threshold, with the caveat that there must be an embedded region $> 25\ \mathrm{dBZ}$ along at least half the axis that is at least $10\ \mathrm{dB}$ greater than the background reflectivity. Level-III reflectivity data has a precision of 5-$\mathrm{dB}$ and will inherently not be able to identify features that are $< 5\ \mathrm{dB}$ different from the background. In general, methods that use fixed thresholds are sensitive to the radar calibration as well as to the grid spacing of the input data since reflectivity values are not scale invariant (Rinehart, 2004).

Several authors have used methods that adapt to changing background reflectivity values in the wider storm and hence better detect localized enhancements than fixed threshold methods. Ganetis et al. (2018) identified both primary and multi-band features by identifying echo regions in NEXRAD Level-II regional reflectivity maps that were greater than the upper-sextile of reflectivity values for a given precipitation region. Ganetis et al. (2018) classified primary bands as objects that were $\geq 200\ \mathrm{km}$ and had an aspect ratio (width/length) of $\leq 0.5$ and multi-bands as objects that were $< 200\ \mathrm{km}$ and had as aspect ratio $\leq 0.5$. Objects that had an aspect ratio $> 0.5$ and a length $\geq 10$ and $\leq 100\ \mathrm{km}$ were labelled as cells. Radford et al. (2019) used NEXRAD base reflectivity (lowest elevation angle) mosaics for three winter seasons and only considered objects that were 1.25 standard deviations above the mean reflectivity, as well as $\geq 250\ \mathrm{km}$ in length and with a minimum aspect ratio of 0.33 following the methods of Baxter and Schumacher (2017).

The Method for Object-Based Diagnostic Evaluation (MODE), included in the Model Evaluation Tools (MET) verification software package, is a popular tool for detecting objects in meteorological datasets (Bullock et al., 2016). Originally developed to compare forecast fields to observed fields, MODE applies a convolution to a field and then uses user-defined thresholds to find objects in the original field. For example, the user can change the size of the convolution radius used to smooth the input field and the single threshold that determines whether an object is defined relative to the smoothed background.

For some applications, detecting only strong snow bands is sufficient. For our research, which aims to understand the environments in which snow bands form and the physical processes that create them, a fuller picture of their life cycle is needed. Visual inspection of sequences of radar data demonstrates that advecting snow bands often undergo transitions from faint to strong to faint before dissipating. In order to study these structures, we needed an automated snow band detection method that would detect a range of echo features from faint to strong.

Our method, described in detail in Section 2, rescales the reflectivity field to an estimated snow rate to better discern weak echo features and combines two differential adaptive thresholds to determine if a feature stands out from the background. Based on the difference between a pixel and the background average value, the algorithm determines if the pixel is part of an enhanced feature. The algorithm "adapts" the difference threshold to the mean background value.There are two ways a pixel could pass this test. One based on a criteria that requires a decreasing difference with increasing background value and one that requires an increasing difference with increasing background value. We use the generic term *locally enhanced feature* to describe objects that one would pick out by eye as distinct from lower background values. We define two varieties of locally enhanced features, *strong* features that have larger differences from the background and *faint* features that have smaller differences from the background where the background field is weak. The algorithm we developed for detecting locally-enhanced features in winter

storms is described in Section 2, examples of our technique are shown in Section 3, and a summary is provided in Section 4. We contributed the software to the open-source Python package, Py-ART (Helmus and Collis, 2016), where it is available for general use. Within this paper, we will be using the terms *object* which is commonly used in the image processing literature, and *feature* which refers here to the meteorological application interchangeably. We also define winter season storms of interest as those that contain a substantial area of surface snow fall.

## 2 Methods

### 2.1 Data

To demonstrate our method, we use NEXRAD Level-II radar reflectivity regional reflectivity maps composed from several radars in the Northeast US (Tomkins et al., 2022, 2023a, b). The regional maps use 2D Cartesian Cressman interpolation to a $2\,\mathrm{km}$ grid based on the $0.5°$ elevation angle from several different radars. Where there is overlap between adjacent radars, we use the point with the highest reflectivity value. Given the coarse vertical spatial resolution of NWS operational radar volume coverage patterns, 3D Cartesian interpolation often smooths and obscures the fine-scale horizontal features we need to discern faint objects. For our application, the varying altitudes along the $0.5°$ elevation angle scans that constitute the regional maps are preferable to a constant altitude map that smooths key features we need for our analysis. While we demonstrate our technique with a specific set of NEXRAD radars in the northeast US, the technique can be applied to any gridded radar data.

### 2.2 Feature detection algorithm

The feature detection method described in this paper to identify locally enhanced reflectivity features in cool-season precipitation systems is built upon the implementation of adaptive thresholds for objective convective-stratiform precipitation classification developed for warm-season storms in a series of papers by Churchill and Houze (1984), Steiner et al. (1995), Yuter and Houze (1997), and Yuter et al. (2005). The underlying idea, identifying the cores of features that exceed the background value by an amount that varies with the background value, is well established (Steiner et al., 1995). These types of algorithms are highly customizable and can be tuned to a wide variety of datasets. So as to be more general purpose, the software we contributed to the open-source python package, Py-ART, can be configured to run either as a variant of established convective-stratiform precipitation algorithm for warm-season storms or for the application described in this paper for winter storms.

A data flow diagram of the winter storm algorithm using the Yourdon symbol conventions (Woodman, 1988) shows the key steps in the data processing (Fig. 1). The top-level data flow (Fig. 1a) is shown with two levels of nested data processing (Fig. 1b and c). The steps in Fig. 1c follow the data flow steps from Steiner et al. (1995) algorithm to identify convective and stratiform precipitation from reflectivity in rain layers. Input parameter names and recommended settings for detecting locally enhanced reflectivity features in snow are provided in Table 1.

The feature detection algorithm outputs 2D arrays that in effect simplify the input reflectivity field into faint feature, strong feature, and background categories. Additional image processing of this output based on the shape characteristics of individual features such as aspect ratio, length, width, and area can be used to further classify the features into different types of banded and cellular features (e.g. Ganetis et al., 2018).

### 2.2.1 Estimation of snow rate

A key difference from previous methods described in Section 1, is the use of an estimated snow rate field as the input for the feature detection instead of a radar reflectivity field. A very rough first order approximation is that radar reflectivity $\mathrm{dBZ} \propto \log_{10}(\mathrm{mass}^3)$ for unrimed aggregates, where mass is the mass per unit volume of precipitation-sized particles (Matrosov et al., 2007). For rain, the radar reflectivity to mass relationship can be approximated by $\mathrm{dBZ} \propto \log_{10}(\mathrm{mass}^2)$. Multiple observational studies have shown that any one relationship between reflectivity and snow rate has high uncertainty since for given $\mathrm{dBZ}$, the associated snow rate can vary by two orders of magnitude (Fujiyoshi et al., 1990; Rasmussen et al., 2003). This first step in the data processing transforms reflectivity to a value that is more linear in liquid equivalent snow rate. We chose to use liquid equivalent snow rate rather than linear Z since it is more physically intuitive. We do not use the derived snow rates for quantitative estimates of precipitation, just as an alternative scaling factor to reflectivity in $\mathrm{dBZ}$.

Empirical Z-S relations encompass ones for dry snow, which have smaller changes of equivalent liquid per $\Delta Z$ to ones for wet snow which have larger changes per $\Delta Z$ (Fig. 2 Rasmussen et al., 2003). In order to obtain higher contrast between locally enhanced Z in terms of snow rates, we use the wet snow Z-S relationship from Rasmussen et al. (2003); $Z_e = 57.3 S^{1.67}$ where $Z_e$ is equivalent radar reflectivity with units of $\mathrm{mm}^6 \ \mathrm{m}^{-3}$ and $S$ is snow rate with units of $\mathrm{mm} \ \mathrm{hr}^{-1}$. Our results are not sensitive to the absolute values of snow rate, only to the relative anomaly from the background average. Examples of re-scaling the reflectivity field to a snow rate field are shown in Fig. 3.

### 2.2.2 Calculation of smoothed background field

A locally smoothed background average snow rate field is computed from the snow rate field (Fig. 3). The background radius smoothing parameter is used to define a circular footprint surrounding each pixel (Fig. 1a). We found that use of circular footprints produced better results than rectangular footprints. The points within the circular footprint are averaged to find the background value for that point. Feature detection is sensitive to the size of the area used to calculate the background value (not shown). We found a background radius of 40 km was the most suitable for detecting snow band features in the NWS NEXRAD data. A larger background radius will yield a smoother background average field used to compare to the input field to find features. A smaller background radius is likely more suitable for warm-season precipitation systems which usually have stronger reflectivity gradients than cool-season precipitation systems. An example of the locally-smoothed background average snow rate field is shown in Fig. 3.

When calculating the background average, a minimum fraction of valid points within the footprint can be set so only pixels with a sufficient amount of surrounding echo are used in the analysis. We use a minimum fraction of 0.75 (i.e. the footprint must contain at least 75% echo coverage to be used in the analysis). This is done to minimize small, spurious features on the

edge of the echo. The effects of the 0.75 minimum fraction can be seen in Fig. 3 where there are differences between the more jagged echo outer edges in the snow rate field (panels d-f) compared to the smoother echo edges in the background field (panels g-i). Changing the minimum fraction acts to change how much echo must be present in the circular background footprint for a

given pixel to be considered in the algorithm. A minimum fraction of zero would yield a background field with identical outer edges to the snow rate field.

### 2.2.3    Two adaptive differential thresholds for finding feature cores

The background average field and the original snow rate field are compared using two difference threshold schemes. Pixels where the difference between the snow rate field and the background average field are greater than or equal to the adaptive

difference threshold constitute a feature's core.

There are two individual pixel versus background difference relationships built into the algorithm, a cosine scheme and a scalar multiplier scheme that are used on combination and utilize units of $\mathrm{mm\ hr^{-1}}$. A pixel is identified as feature core if the value of the pixel exceeds the background by either adaptive threshold. If the pixel is only identified as a core with the scalar multiplier scheme, it is labeled as a faint feature. If it is identified as a core with the cosine scheme it is labeled as a strong

feature. The cosine relationship has a decreasing threshold with increasing background value (Fig. 4a). The cosine scheme uses simple and intuitive parameters to define a smooth, curved relationship between the background value and the difference threshold. The scalar multiplier scheme uses a difference threshold that increases linearly as the background value increases (Fig. 4b). This allows the scalar multiplier to pick up subtle features that are not very distinct from the background when the background values are small (i.e. in regions of weak precipitation). For example, for a background value of $1\ \mathrm{mm\ hr^{-1}}$, the

cosine scheme threshold is $1.4\ \mathrm{mm\ hr^{-1}}$ while the scalar scheme threshold is $0.5\ \mathrm{mm\ hr^{-1}}$. After extensive testing on many idealized and real examples from winter storms, we found that a combination of both types of adaptive thresholds was needed in order to detect the full range of reflectivity features from faint to strong. The cosine scheme only identifies objects that are very distinct from the background, while the scalar multiplier scheme identifies objects that are both very distinct and not very distinct. We chose the particular equations described here as they were both intuitive and easy to tune.

The cosine scheme's decreasing difference threshold with increasing background value is described in Equation 1 where $S$ represents the snow rate at a pixel, $S_{average}$ represents the background average snow rate, $a$ represents a maximum possible difference value corresponding when the background average value is $0\ \mathrm{mm\ hr^{-1}}$ and $b$ represents the background average value where the corresponding difference threshold is zero.

$$\mathrm{S} - \mathrm{S}_{average} \geq a\cos\left(\frac{\pi \mathrm{S}_{average}}{2b}\right) \tag{1}$$

Other similar equations with a decreasing threshold with increasing background value would also likely be suitable. The cosine scheme (Fig. 4a) is adapted from methods used to identify convective and stratiform precipitation structures in rain (e.g. Steiner et al., 1995; Yuter and Houze, 1997; Yuter et al., 2005; Powell et al., 2016). The choice of this specific equation is purposeful as it permits the same Python code to be used with an input field of radar reflectivity from a rain layer and appropriate parameter settings to exactly reproduce the data processing of the original C++ code used in Yuter et al. (2005).

Figure 4a shows how changing the maximum difference ($a$ in Eqn. 1; horizontal dashed line) and zero difference cosine value ($b$ in Eqn. 1; vertical dashed line where the function would cross the x-axis) changes the overall shape of the difference function and thus the thresholds used to identify pixels that are cores. Having a lower maximum difference or zero difference cosine value will increase the number of cores since it relaxes the difference threshold needed for a point to be considered a core. The final tuning parameter in the difference relationship is the "always core threshold" which is the value above which all background points are considered cores (vertical dashed line in Fig. 4). An absolute threshold like the always core threshold is helpful for identifying cores in regions where the background values are high. It is particularly useful for the scalar scheme and provides additional flexibility in turning. The zero difference cosine value can be used in place of the always core threshold in the cosine scheme. In our method, the value corresponding to a pixel that is always part of a snow band is set at an equivalent liquid precipitation rate of 5 mm hr$^{-1}$ (which corresponds to a reflectivity value of 30 dBZ). For reflectivity fields in rain, usually this value is set at or above 40 dBZ (rain rate of about 13 mm hr$^{-1}$).

The scalar multiplier scheme uses a linear function with a difference threshold that increases with increasing background value up to the "always core threshold" (Fig. 4b). The equation for the scalar multiplier scheme is described by Eqn. 2 where $S$ represents the snow rate at a pixel, $S_{average}$ represents the background average snow rate, $c$ represents the scalar difference.

$$\text{S} - \text{S}_{average} \geq (c * \text{S}_{average}) - \text{S}_{average} \tag{2}$$

The scalar difference value ($c$ in Eqn. 2) changes the slope of the difference threshold in Fig. 4b. A larger scalar difference value will yield a steeper slope and a greater difference threshold needed for a given background average value.

Figure 4c shows both difference equations and is colored coded by classification (strong feature, faint feature, background) based on the two different schemes.

A detection threshold that increases with increasing background value helps to distinguish both the tapered edges of stronger features as well as features that differ only slightly from the background. Figure 5 shows three examples of the output from both the cosine scheme and the scalar scheme. Both the cosine scheme and the scalar scheme pick up the strong features from the snow rate (e.g. band of 10+ mm hr$^{-1}$ in Fig. 5b), but only the scalar scheme can identify the weaker features including the fuzzy, irregular edges.

Examples in appendix A demonstrate the influence of the background radius, always core threshold, scalar difference, zero difference cosine value, and maximum difference on the algorithm output.

### 2.2.4 Converting cores to contiguous features

To address isolated pixels within detected features, we perform a binary closing on the 2D array of cores to mitigate these artifacts (Fig. 6a). A binary closing is an image dilation followed by an image erosion which acts to fill in the holes within a feature but keeps the feature at roughly the original size (Jamil et al., 2008). We use a quasi-circular 5x5 kernel (Fig. 6b) for the binary closing to yield a more physically realistic output as opposed to use of a square kernel.

After we perform the binary closing step, we then remove objects that are less than 120 km$^2$ in area. We found that this value was suitable for our applications. No object capable of meeting the band criteria of Ganetis et al. (2018) is less than 120 km$^2$

in area. An example of the binary closing and small object removal on the cosine scheme cores from the examples presented in Fig. 5 is shown in Fig. 7 to yield the filtered, spatially contiguous features of interest.

There were two steps from the established convective-stratiform algorithm that we turned off for our feature detection application to winter storms. An additional step can be applied to delineate a weaker echo subset of the background echo. We do not use this for our application and set both the weak echo and minimum value to $0 \text{ mm hr}^{-1}$ (Table 1). Alternate values of these settings can be useful for tabulating statistics of different magnitudes of background radar echo. For the radar data set we were using, we found that the additional step of use a radius of influence around each core pixel as part of the feature was not needed. To turn this off, we set the maximum core radius to 2 km, the same as the input grid pixel size (Table 1). For some applications, the radius of influence step may be needed, especially for finer grids.

### 2.2.5 Snow storm faint and strong feature identification method

Objects that are identified by the cosine scheme we define as "strong" objects, while objects that are only identified by the scalar multiplier and not by the cosine scheme are defined as "faint" objects (Fig. 8). The separation into strong and faint objects allows for analysis that addresses the relative intensity of the observed reflectivity compared to independent data sets such as surface weather station snow rates. The output of the algorithm can yield strong and faint portions of the same contiguous feature as well as objects that are solely of one type (Figure 8ghi).

An important component of running the algorithm in practice is to account for uncertainties in the observed data and that no one method for feature detection will work perfectly in all situations. Similar to Yuter et al. (2005), we bound our feature identification by running the algorithm on the estimated snow rate field and two offsets of that field with slightly higher and lower values to yield purposeful over and under estimates of the feature detection. Increasing the radar reflectivity by 2 dB, converting to snow rate, and then running the algorithm yields an overestimate in feature area, while decreasing by 2 dB yields an underestimate. For the underestimate, since we do not consider values $\leq 0 \text{ mm hr}^{-1}$ (Table 1), echo where the original reflectivity field $\leq 2$ dBZ gets removed, so the underestimate feature detection field will have less total echo area than the best and overestimate feature detection fields. Bounding the best estimate feature detection field can be accomplished by varying the input field slightly as we have done here, or by varying the difference equation. Both accomplish the same goal of making minor adjustments to yield an under and over estimate in the field. We recommend adjusting the field by at least $\pm$ 2 dB as this value is close to the minimum uncertainty in the US NWS operational radar reflectivity calibrations. As compared to the "best estimate", the underestimate version usually reduces the size of strong features and amplifies the detection of faint features compared to the best estimate. The overestimate version the snow field usually yields larger feature sizes for the strong features and damps the detection of the faint features compared to the best estimate.

### 2.3 Visual de-emphasis of regions with mixed precipitation

After we run the algorithm to detect features, we apply image muting (Tomkins et al., 2022) as a separate step independent of the feature detection algorithm to identify regions of mixed precipitation in the winter storms. This step de-emphasizes portions of the echo that pass through the 0°C level in the final visualized plots by utilizing information from the radar's correlation

coefficient field. Regions where the reflectivity is $\geq 20$ dBZ and the correlation coefficient are $\leq 0.97$ are considered to be likely melting or mixed precipitation and are colored in a grayscale (Tomkins et al., 2022). The sharp temperature gradients in winter storms can yield mixed precipitation echo regions that resemble bands (e.g. Picca et al. (2014) their Fig. 2 and Colle et al. (2023) their Fig. 7). It is important to remove these mixed phase echoes before interpreting the detected enhanced features as snow. Full details of how the image muting is applied and evaluated can be found in Tomkins et al. (2022).

## 2.4 Limitations

The quality of the input data fields, including their calibration and precision, are constraints that impact the quality of the output. For example, this particular algorithm would not work well on input radar reflectivity data with 5-dB precision. Input data quality issues such as attenuation are not able to be corrected inside the algorithm. As expected, when we applied our algorithm to Ku-band and Ka-band radar data collected from aircraft we found that attenuation can affect the detection of enhanced features. While the algorithm runs fairly fast (about 60 seconds per 601 x 601 size reflectivity input field on our servers), there is room for improvement in code efficiency and the potential for parallelization to be incorporated.

An important consideration when interpreting the feature detection field is situational awareness of where snow is occurring. Winter storms in the northeast US often transition among multiple precipitation types including snow, rain, mixed, and melting precipitation. We tuned the algorithm to work specifically in regions where we were reasonably confident that surface snow was occurring. Because regions of rain typically have high reflectivities, they are almost always identified as enhanced features based on how we tuned our algorithm to detect features in snow. Our image muting technique (Tomkins et al., 2022) assists with the interpretation of precipitation type within echo by identifying regions of likely mixed and melting precipitation using the correlation coefficient field. Additional, independent datasets such as surface air temperatures, and surface-based precipitation type sensors can also help provide context for the precipitation type observed by radar.

The *flashing* of features occurs when a particular enhanced feature alternates between being detected and not being detected in sequential times. A key goal of the algorithm development was to minimize flashing of individual features in consecutive times. Small speckles are more prone to flashing than larger area features which is why we filter out small objects. While we minimized flashing as best we could, there are still times when features are not consistent through time. Another aspect of flashing occurs where the edges of enhanced features can alternate between strong and faint.

With the *turning knobs* on the command line, it is straightforward to test and to refine the input parameters for the algorithm (also see Appendix A). There is no one set of input parameters that will work perfectly every time. For best results, the user needs to optimize the parameters for their application. It is recommended to use a tuning data set of test cases representing a wide range of possible input fields including time sequences from a dozen or more real cases. Like an out of tune piano, poor tuning will obviously lead to poor results. The input parameters in Table 1 were tuned for the US NEXRAD network of S-band radars and mid latitude northeast US winter storms. These particular values will likely need to be modified for other radar hardware and/or other storm regimes such as polar winter storms.

## 3 Examples

We illustrate our algorithm on regional composites of Level-II data from National Weather Service (NWS) Next-Generation Radar (NEXRAD) network radars that were obtained from the NOAA Archive on Amazon Web Services (Ansari et al., 2018). Full details on how the composites are created can be found in Section 2.1 of Tomkins et al. (2022).

Our examples span a range of cases and snow band intensities including storms with and without primary bands and multi-bands. The example from 7 February 2021 (left panel in Fig. 3, 5, 7, and 8) shows several strong bands over Maryland and Virginia and a few faint objects as well. The one from 17 December 2020 (middle panel in Fig. 3, 5, 7, and 8) includes a strong primary band over northern Pennsylvania and southern New York and several faint bands over southern Pennsylvania. The data from 17 December 2019 (right panel in Fig. 3, 5, 7, and 8) contains mostly faint bands over New Hampshire and Maine. All the cases shown also include portions of echo that contain mixed precipitation which commonly occurs in east coast US winter storms. In each of the example regional cases, video supplements illustrate the time continuity of the detection method as features evolve and move through the domain.

### 3.1 Application in snow layers to identify faint and strong reflectivity features

The spatial and temporal coherence of the bands is illustrated in the sequences of images $\pm$ 1 hour for each of Fig. 9, 10, 11, 12 in the Video Supplement. Individual bands form and dissipate as the storm moves and evolves.

The winter storm from 7 February 2021 at 14:37 UTC exhibited a primary band extending from northern Virginia to Connecticut and faint multi-bands across Pennsylvania and New York (Fig. 9). The underestimate field (Fig. 9c) also has a strong primary band similar to the best estimate although smaller and narrower. The few, small strong features in the best estimate are detected as faint features in the underestimate (Fig. 9c). The overestimate field (Fig. 9d) has a wider strong band compared to the best estimate and has more strong objects in general compared to the best estimate. The strong, primary band traverses along the east coast while the faint multi-bands dissipate and form in the weaker region in Pennsylvania and New York. (Video Supplement Animation-Figure-9).

The winter storm from 17 December 2020 over the Northeast US (Fig. 10) contained primary and multi-bands. There are several large bands that extend over New York and Massachusetts that are associated with high values in the snow rate field and are identified as strong features (Fig. 10). Over southern Pennsylvania there are other features that do not stand out as much that are identified as faint features (Fig. 10). As the storm evolves, the large band remains roughly in the same location but changes shape while the other, smaller features undergo more dramatic changes (e.g. dissipate, break apart, strengthen) (Video Supplement Animation-Figure-10). The faint bands over Pennsylvania also evolve in time and space, some transitioning to strong bands and some weakening and dissipating (Video Supplement Animation-Figure-10). Similar to the previous example, the under estimate has a narrower primary band and has a lot more "faint" features compared to the best and over estimates. The over estimate shows very few faint features and mostly amplifies the main strong features (Fig. 10d).

The winter storm on 17 December 2019 was generally weaker and had a lot of faint bands compared to the example from 17 December 2019 (Fig. 11). Areas of the southern part of the storm are image muted, indicating melting and mixed precipitation

and a transition to rain. The northern part of the storm has numerous faint features over northern New York, Vermont, New Hampshire, and Maine (Fig. 11a,b). In this example, the faint bands are more coherent in time and space than the other examples and some of these faint bands evolve into strong bands and some strong bands evolve into faint bands (Video Supplement Animation-Figure-11).

The winter storm from 7 February 2020 over the northeast US that is characterized by large regions of melting (grey muted regions in Fig. 12). This example has a large, strong object extending from Pennsylvania through New York but does not have any faint bands or sets of multi-banded structures as discussed in Colle et al. (2023) (Fig. 12). This large, long, strong feature spans the transition from snow to rain and is persistent in time for several hours (Video Supplement Animation-Figure-12). It is very likely that the portion of the strong band to the east of the SW-NW mixed precipitation area is rain rather than snow. Further feature filtering by surface air temperature fields would be useful in cases like this to isolate surface snow.

## 4 Summary

We present a novel method for identifying locally-enhanced features in radar observations of winter storms that uses a combination of increasing and decreasing adaptive thresholds as a function of average background values. Our method identifies features from a snow rate field rather than radar reflectivity in order to better automatically identify human eye discernable features in radar data of snow. Previous methods to automatically detect snow bands in radar observations either used inflexible thresholds and Level-III reflectivity data (5-dB precision) or used adaptive thresholds that were not able to detect objects that are not very distinct from the background. This new method facilitates both the detection of stronger objects and fainter objects that are less distinct from the background average in snow storms. The wider range of characteristics of detected features provides a more comprehensive basis for examining hypotheses relating radar-observed features to surface snowfall and intra-storm environments.

There is no one feature detection algorithm that is going to produce perfect results every time. Our image processing software facilitates easy adjustments to the algorithm tuning from the command line and has built functionality for determining credible under and overestimates of feature areas. These under and overestimates aid in bounding the uncertainty in the feature detection field. The user is advised to always test and to refine the tuning parameters of the algorithm on their data to ensure that the settings are adequate for their purpose.

The output of the algorithm described in this paper yields 2D arrays with categorical values for different strengths of detected radar echo features and background echo. These output arrays can be input into image processing software to yield statistics of feature characteristics such as area, aspect ratio, orientation, convex hull, centroid location, etc. Object attributes can be used to further subset objects and for comparison to other independent data sources. Additionally, this algorithm can be applied to snow rate fields from numerical forecast model output to yield feature objects for use in nowcasting and for model evaluation.

Differential adaptive threshold methods for image segmentation that distinguish locally enhanced features from a varying background have applications to several areas in geosciences. In satellite data analysis, detection of cold cloud tops associated with deep convective storm anvils is often defined based on absolute IR brightness thresholds (Schiffer and Rossow, 1983;

Arkin and Meisner, 1987; Machado and Rossow, 1993) but tropopause heights can vary latitudinally, regionally, and seasonally.

Additionally, satellite passive microwave brightness temperature signatures associated with local enhancements in scattering and emission by precipitation are harder to discern over the the more spatially varying thermal characteristics of land as compared to ocean (Ferraro et al., 2013).

*Code and data availability.* Data: The NWS NEXRAD Level-II data used in Figs. 9, 10, 11, and 12 can be accessed from the National Centers for Environmental Information (NCEI) at https://www.ncei.noaa.gov/products/radar/next-generation-weather-radar. The

325 radar composites created from the NEXRAD Level-II data used in Figs. 9, 10, 11, and 12 can be accessed from a Dryad repository at https://doi.org/10.5061/dryad.rbnzs7hj9.

Code: We submitted functions that run the feature detection algorithm to the Py-ART GitHub repository (Helmus and Collis, 2016) to facilitate use of this technique by others. They were accepted and released in Py-ART version 1.14.1. The Py-ART function used to create the figures in the paper can be accessed via https://arm-doe.github.io/pyart/API/generated/pyart.retrieve.feature_detection.html. An example

of how to use the function is provided here: https://arm-doe.github.io/pyart/examples/retrieve/plot_feature_detection.html.

*Video supplement.* List of animations with captions and filenames

All animations can be viewed at: https://av.tib.eu/series/1524/. Individual animations can be viewed by following the DOI URL.

Animation-Figure-9: Animated plot of Fig. 9 demonstrating bounding the best estimate feature detection with purposeful

overestimates and underestimates using an example from 7 February 2021 13:30-15:30 UTC which features a primary snow band and a few multi-bands. Locally enhanced features that include mixed precipitation are image muted in gray (Tomkins et al., 2022). (a) Re-scaled snow rate field ($mm\ hr^{-1}$ units), Feature detection (b) best estimate, (c) underestimate, (d) overestimate. Feature detection fields show background regions in teal, strong features in yellow, and faint features in orange.

Title: 07 February 2021 feature detection example DOI: http://doi.org/10.5446/63170

Animation-Figure-10: Animated plot of Fig. 10 demonstrating bounding the best estimate feature detection with purposeful overestimates and underestimates using an example from 17 December 2020 05:30-07:30 UTC which features several strong primary bands and a few faint multi-bands. Locally enhanced features that include mixed precipitation are image muted in gray. (a) Re-scaled snow rate field ($mm\ hr^{-1}$ units), Feature detection (b) best estimate, (c) underestimate, (d) overestimate. Feature detection fields show background regions in teal, strong features in yellow, and faint features in orange.

Title: 17 December 2020 feature detection example DOI: http://doi.org/10.5446/63171

Animation-Figure-11: Animated plot of Fig. 11 demonstrating bounding the best estimate feature detection with purposeful overestimates and underestimates using an example from 17 December 2019 15:30-17:30 UTC which features many faint multi-bands. Locally enhanced features that include mixed precipitation are image muted in gray. (a) Re-scaled snow rate

field (mm hr$^{-1}$ units), Feature detection (b) best estimate, (c) underestimate, (d) overestimate. Feature detection fields show background regions in teal, strong features in yellow, and faint features in orange.

Title: 17 December 2019 feature detection example DOI: http://doi.org/10.5446/63172

Animation-Figure-12: Animated plot of Fig. 12 demonstrating bounding the best estimate feature detection with purposeful overestimates and underestimates using an example from 7 February 2020 12:30-14:30 UTC which features a large primary band, portions of which are mixed precipitation and image muted in gray. (a) Re-scaled snow rate field (mm hr$^{-1}$ units), Feature detection (b) best estimate, (c) underestimate, (d) overestimate. Feature detection fields show background regions in teal, strong features in yellow, and faint features in orange.

Title: 7 February 2020 feature detection example DOI: http://doi.org/10.5446/63168

## Appendix A: Impact of tuning parameters

By design, adjusting the tuning parameters in the algorithm changes the detected features in the output. To demonstrate how each parameter influences the output, we run the algorithm with many different configurations on an example from Long Island, NY in February 2021 (Fig. A1). Example output of the scalar scheme portion of the algorithm and how it varies with different combinations of scalar difference (1.2x, 1.5x, and 1.8x), background radius (20, 40 and 60 km), and always core threshold (4, 5 and 6 mm hr$^{-1}$) is shown in Figs. A2, A3, and A4. Example output of the cosine scheme portion of the algorithm and how it varies with different combinations of zero difference cosine value (4, 5, and 6 mm hr$^{-1}$), maximum difference (0.5, 1.5, and 2.5 mm hr$^{-1}$), and always core threshold (4, 5, and 6 mm hr$^{-1}$) is shown in Figs. A5, A6, and A7. As expected, a lower scalar difference (1.2x, Fig. A2) picks up more distinct features and larger areas for a given feature compared to a higher scalar difference (1.8x, Fig. A4) since a lower scalar difference setting will result in a smaller difference needed for a feature to be identified from the background (Fig. 4b). In general, fewer separate features with larger areas are identified when the background radius is larger compared to when the background radius is smaller (compare across a row from left to right in Figs. A2–A4). As the always core threshold setting is increased from 4 mm hr$^{-1}$ to 6 mm hr$^{-1}$ the areas of detected features decrease as the weaker values along the tapered edges of the local enhancements fall below the detection threshold (compare down a column from top to bottom in Figs. A2–A7). Similar to the always core threshold, the areas of the detected features decrease as the zero difference cosine value increases from 4 mm hr$^{-1}$ to 6 mm hr$^{-1}$ as the tapered edges of the features no longer meet the detection threshold (Figs. A5–A7). A lower maximum difference value decreases the threshold needed to identify cores when the background value is low. As the maximum difference value increases, less echo is identified as a feature, particularly along the edges of the locally-enhanced features (compare across a row from left to right in Figs. A5–A7). Depending on the user's specific application, any one of these 54 variations may be most suitable. A key to this algorithm is its built-in flexibility.

*Author contributions.* LMT and SEY conceptualized the project and designed the methodology with input from MAM. LMT wrote the Python software with input from SEY and MAM. LMT prepared the manuscript and the figures. All authors contributed to editing and review.

*Competing interests.* The authors declare that they have no conflict of interest.

*Acknowledgements.* Our testing methodology and iterative algorithm refinement benefited from discussions with Brian Colle, Phillip Yeh, Luke Allen, and Kevin Burris. This research has been supported by the National Science Foundation (AGS-1905736), the National Aeronautics and Space Administration (80NSSC19K0354), and the Center for Geospatial Analytics at North Carolina State University.

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

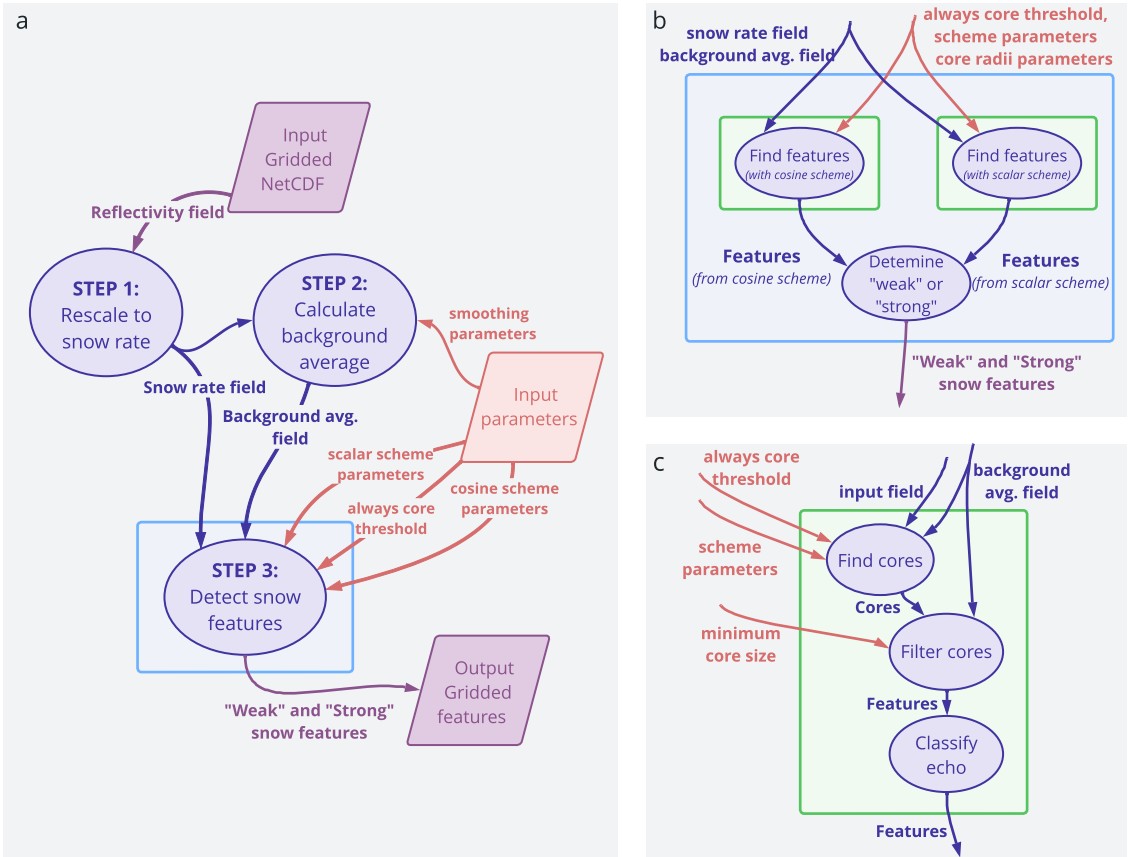

**Figure 1.** Data flow diagram of the winter storm feature detection algorithm. Dark blue ovals indicate processes, purple polygons indicate input and output data, and orange elements indicate adjustable setting parameters used in the functions. Each arrow represents an input or output to the associated functions. The reflectivity field and background average field are 2D arrays. The strong and faint snow features are represented by distinct values in a 2D array. a) Top level data flows. b) The detailed steps within "Step 3: Detect snow features" (blue) box in panel a). c) shows the detailed steps within the "Find features" (green) boxes in panel b).

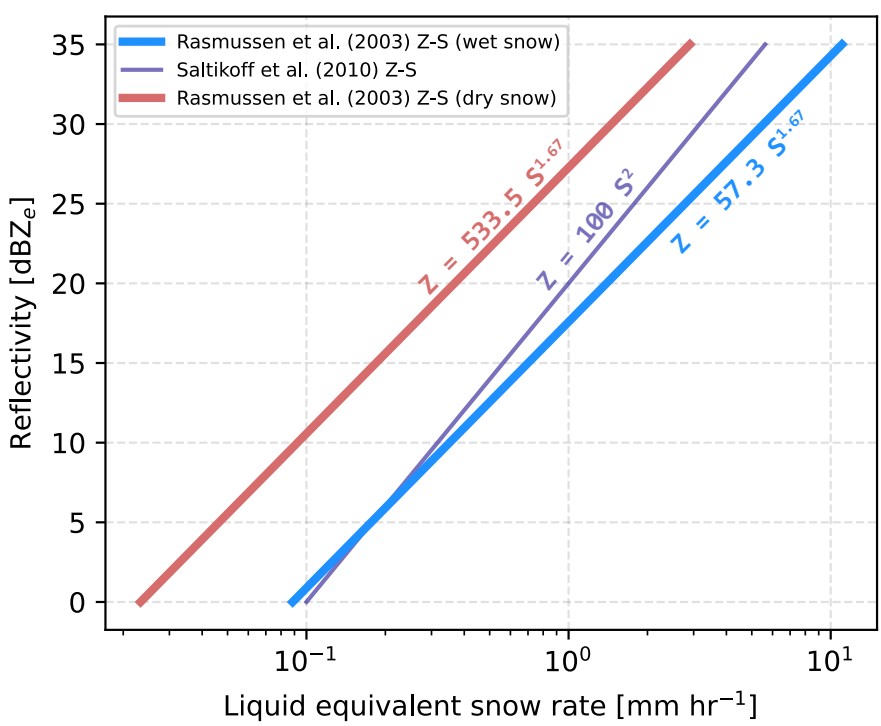

**Figure 2.** Reflectivity to snow rate (Z-S) relationships with log-scale x-axis. The bold blue line indicates the relationship from Rasmussen et al. (2003) for wet snow used in this study. The bold red line shows the relationship for dry snow from Rasmussen et al. (2003) and the purple line shows the relationship for snow used by the Finnish Meteorological Institute's Doppler radar network (Saltikoff et al., 2010).

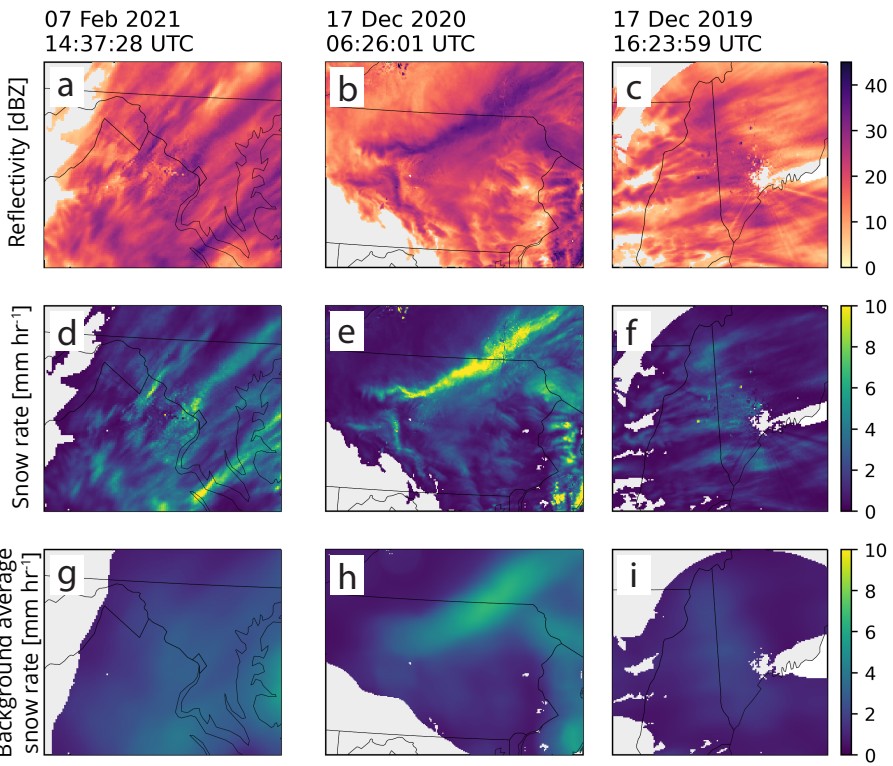

**Figure 3.** Close-up examples of (a)-(c) Reflectivity [dBZ] rescaled to (d)-(f) snow rate [mm hr$^{-1}$] and smoothing of the snow rate fields to a (g)-(i) background average using a 40 km radius footprint. (a), (d), (g) from 7 February 2021 14:37:28 UTC with an area of 263 km x 222 km, (b), (e), (h) from 17 December 2020 16:26:01 UTC with an area of 472 km x 361 km, and (c), (f), (i) from 17 December 2019 16:23:59 UTC with an area of 326 km x 306 km. Grid spacing is 2 km x 2 km for all examples.

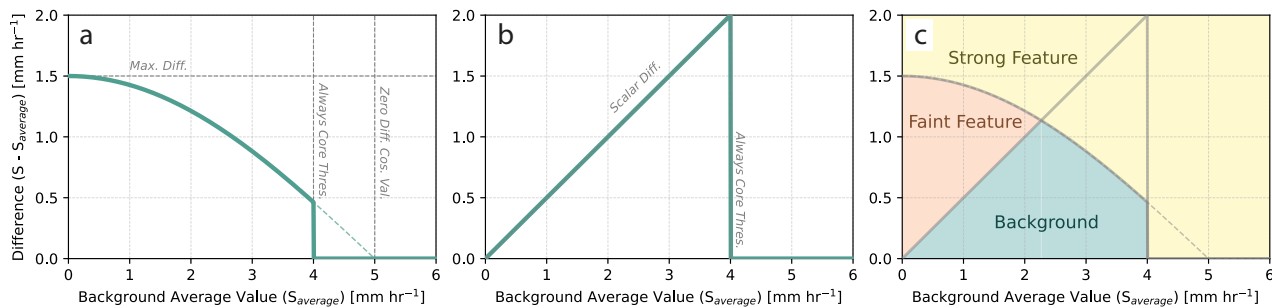

**Figure 4.** Adaptive difference relationships used to determine the threshold between a pixel and its background value to designate the pixel as a feature core. (a) cosine scheme and (b) scalar multiplier scheme. Panel (c) shows the difference relationships in (a) and (b) and is shaded based on the where each feature type is found. Note y-axis range in (b) extends further than in (a) and (c). Input parameters used for tuning are annotated with gray dashed lines, see text for full details.

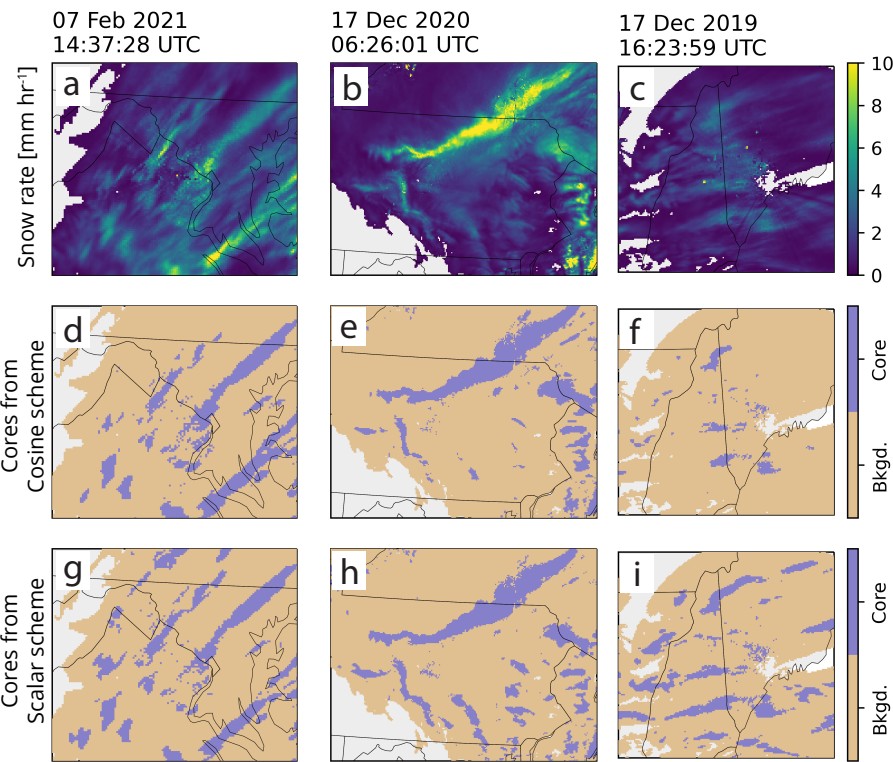

**Figure 5.** Close-up examples of (a)-(c) snow rate [mm hr$^{-1}$], (d)-(f) feature cores detected with the cosine scheme, and (g)-(i) feature cores detected with the scalar scheme. (a), (d), (g) from 7 February 2021 14:37:28 UTC with an area of 263 km x 222 km, (b), (e), (h) from 17 December 2020 16:26:01 UTC with an area of 472 km x 361 km, and (c), (f), (i) from 17 December 2019 16:23:59 UTC with an area of 326 km x 306 km. Grid spacing is 2 km x 2 km for all examples.

a) Binary closing operation

b) Kernel used for operation

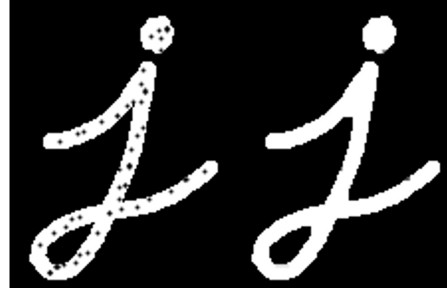

| 0 | 0 | 1 | 0 | 0 |
|---|---|---|---|---|
| 0 | 1 | 1 | 1 | 0 |
| 1 | 1 | 1 | 1 | 1 |
| 0 | 1 | 1 | 1 | 0 |
| 0 | 0 | 1 | 0 | 0 |

**Figure 6.** (a) Example of binary closing operation (image morphology dilation then erosion) from https://docs.opencv.org/4.x/d9/d61/tutorial_py_morphological_ops.html and (b) kernel used in binary closing operations.

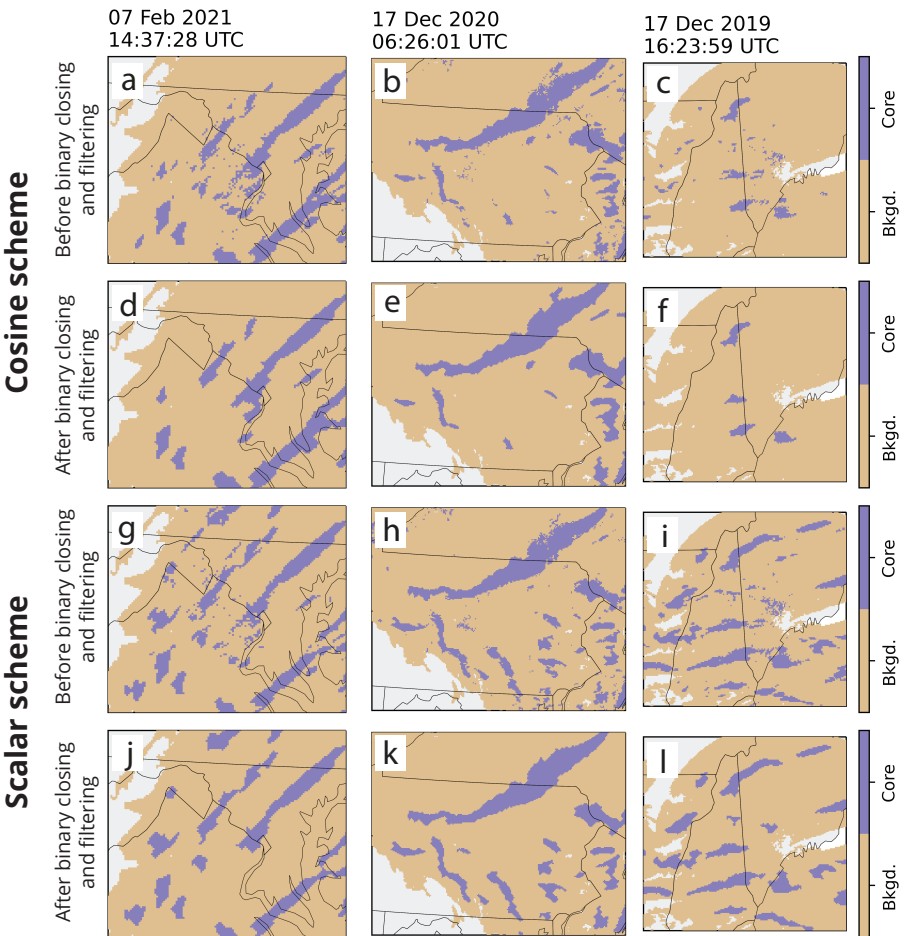

**Figure 7.** Close-up examples of feature cores from cosine scheme (a)–(f) and scalar scheme (g)–(l) before and after binary closing and removal of small objects. The bottom row represents the filtered cores. (a), (d), (g), (j) from 7 February 2021 14:37:28 UTC with an area of 263 km x 222 km, (b), (e), (h), (k) from 17 December 2020 16:26:01 UTC with an area of 472 km x 361 km, and (c), (f), (i), (l) from 17 December 2019 16:23:59 UTC with an area of 326 km x 306 km. Grid spacing is 2 km x 2 km for all examples.

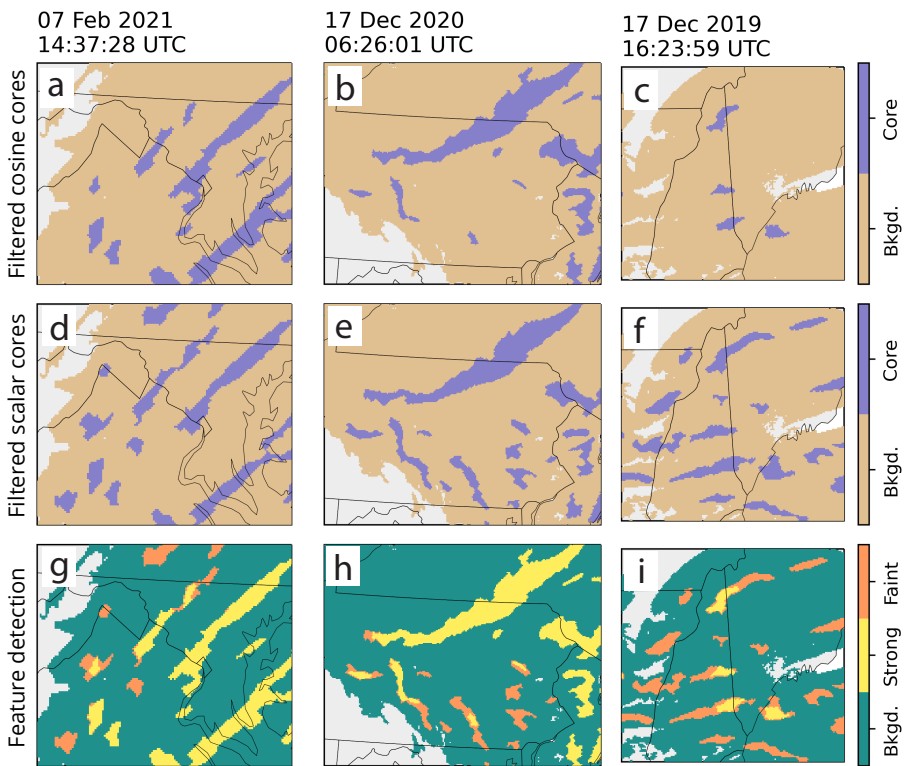

**Figure 8.** Close-up examples of (a)-(c) filtered features from cosine scheme, (d)-(f) filtered features from scalar scheme, and (g)-(i) feature detection output wherein portions of objects labeled as strong were detected in the cosine scheme and those labeled faint are only detected in the scalar scheme. (a), (d), (g) from 7 February 2021 14:37:28 UTC with an area of 263 km x 222 km, (b), (e), (h) from 17 December 2020 16:26:01 UTC with an area of 472 km x 361 km, and (c), (f), (i) from 17 December 2019 16:23:59 UTC with an area of 326 km x 306 km. Grid spacing is 2 km x 2 km for all examples.

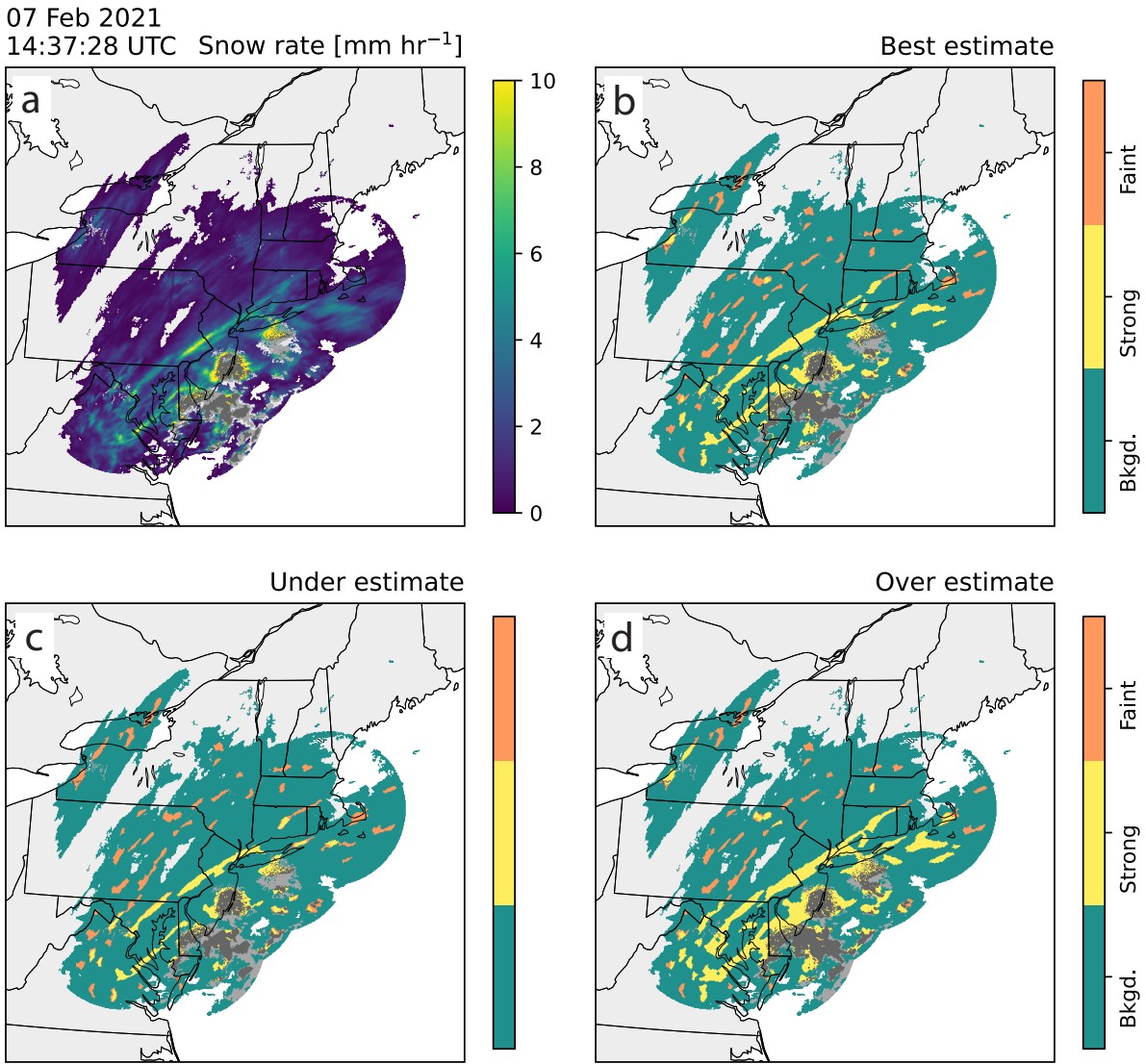

**Figure 9.** Demonstration of bounding the best estimate feature detection with purposeful overestimates and underestimates using an example from 7 February 2021 14:37:28 UTC which features a primary snow band and a few multi-bands. Locally enhanced features that include mixed precipitation are image muted in gray (Tomkins et al., 2022). Area of map is 1202 km x 1202 km and grid spacing is 2 km x 2 km. (a) Re-scaled snow rate field ($\mathrm{mm\ hr^{-1}}$ units), Feature detection (b) best estimate, (c) underestimate, (d) overestimate. Feature detection fields show background regions in teal, strong features in yellow, and faint features in orange. An animated version of this figure is available in the Video Supplement Animation-Figure-9.

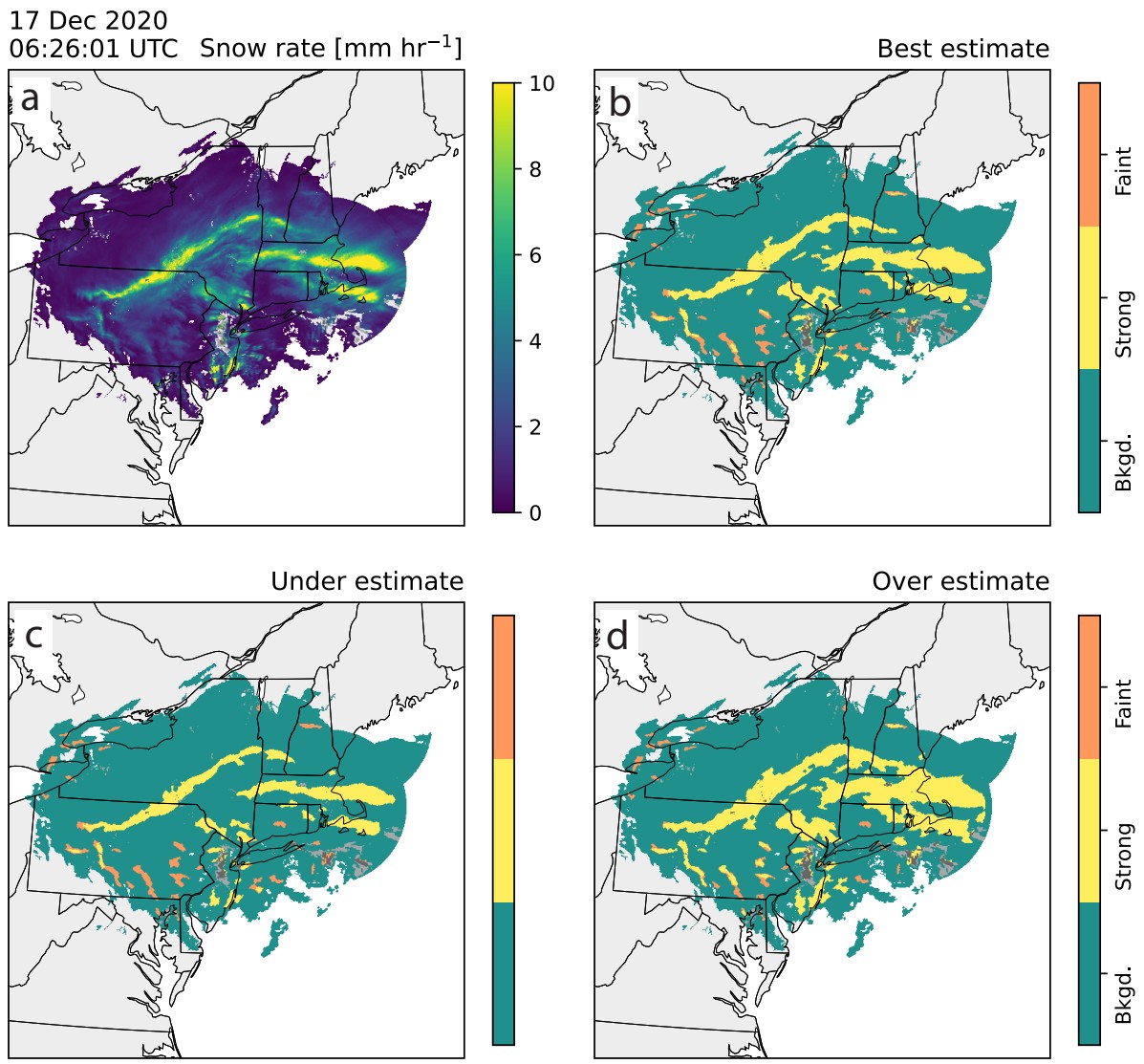

**Figure 10.** Demonstration of bounding the best estimate feature detection with purposeful overestimates and underestimates using an example from 17 December 2020 06:26:01 UTC which features several strong primary bands and a few faint multi-bands. Locally enhanced features that include mixed precipitation are image muted in gray. Area of map is 1202 km x 1202 km and grid spacing is 2 km x 2 km. (a) Re-scaled snow rate field ($\mathrm{mm\ hr^{-1}}$ units), Feature detection (b) best estimate, (c) underestimate, (d) overestimate. Feature detection fields show background regions in teal, strong features in yellow, and faint features in orange. An animated version of this figure is available in the Video Supplement Animation-Figure-10.

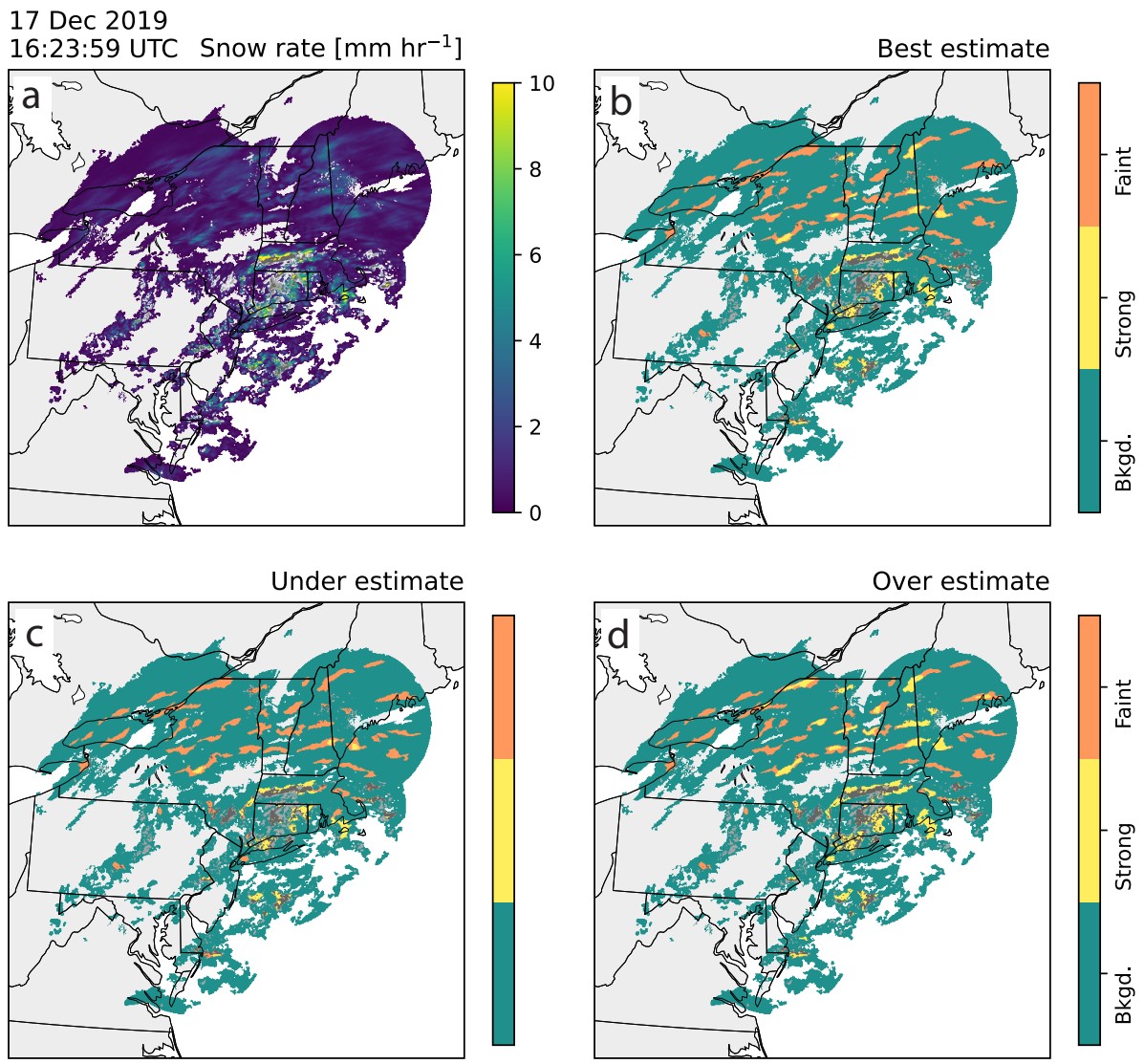

**Figure 11.** Demonstration of bounding the best estimate feature detection with purposeful overestimates and underestimates using radar example from 17 December 2019 16:23:59 UTC which features many faint multi-bands. Locally enhanced features that include mixed precipitation are image muted in gray. Area of map is 1202 km x 1202 km and grid spacing is 2 km x 2 km. (a) Re-scaled snow rate field (mm hr$^{-1}$ units), Feature detection (b) best estimate, (c) underestimate, (d) overestimate. Feature detection fields show background regions in teal, strong features in yellow, and faint features in orange. An animated version of this figure is available in the Video Supplement Animation-Figure-11.

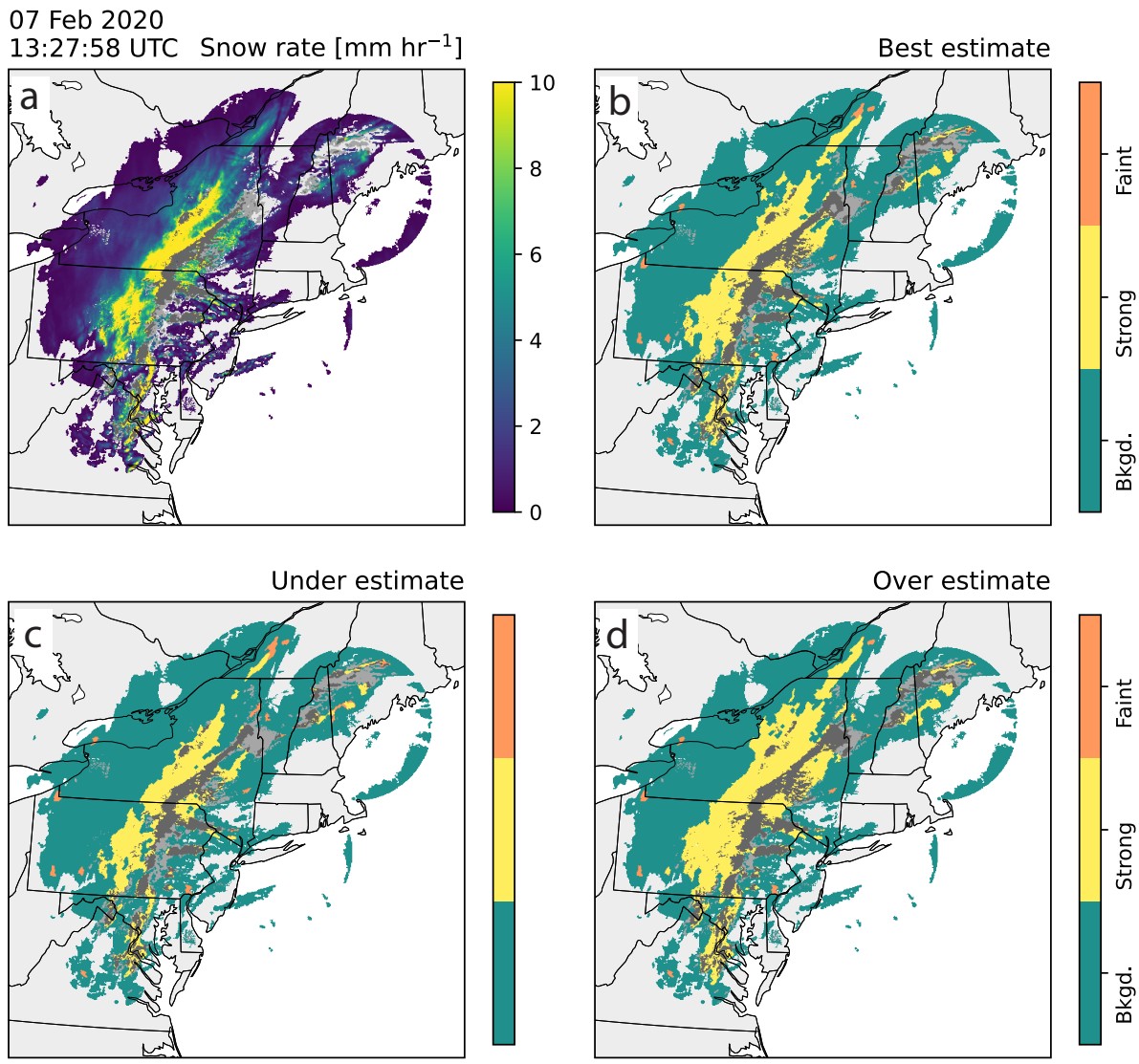

**Figure 12.** Demonstration of bounding the best estimate feature detection with purposeful overestimates and underestimates using radar example from 7 February 2020 13:27:58 UTC which features a large primary band, portions of which are mixed precipitation and image muted in gray. Area of map is 1202 km x 1202 km and grid spacing is 2 km x 2 km. (a) Re-scaled snow rate field (mm hr$^{-1}$ units), Feature detection (b) best estimate, (c) underestimate, (d) overestimate. Feature detection fields show background regions in teal, strong features in yellow, and faint features in orange. An animated version of this figure is available in the Video Supplement Animation-Figure-12.

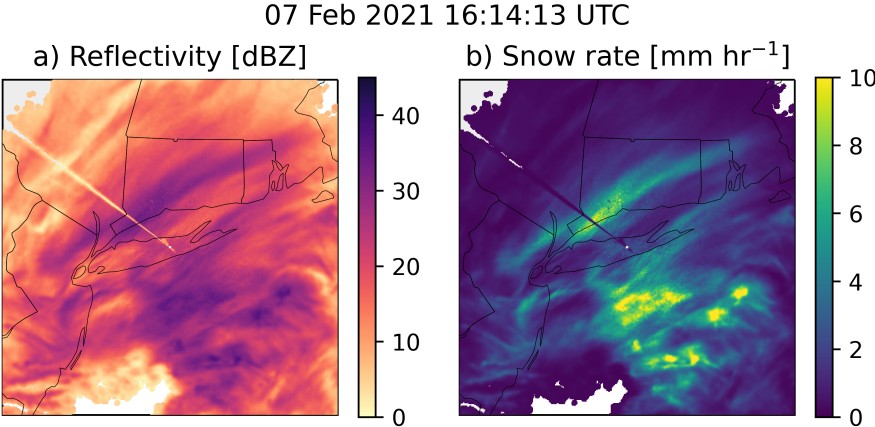

**Figure A1.** (a) Reflectivity [dBZ] and (b) snow rate [mm hr$^{-1}$] from the Long Island, NY NEXRAD radar (KOKX) for 7 February 2021 16:14:13 UTC. Data is shown for a 401 km x 401 km grid with 2 km grid spacing.

## Scalar Difference = **1.2x**

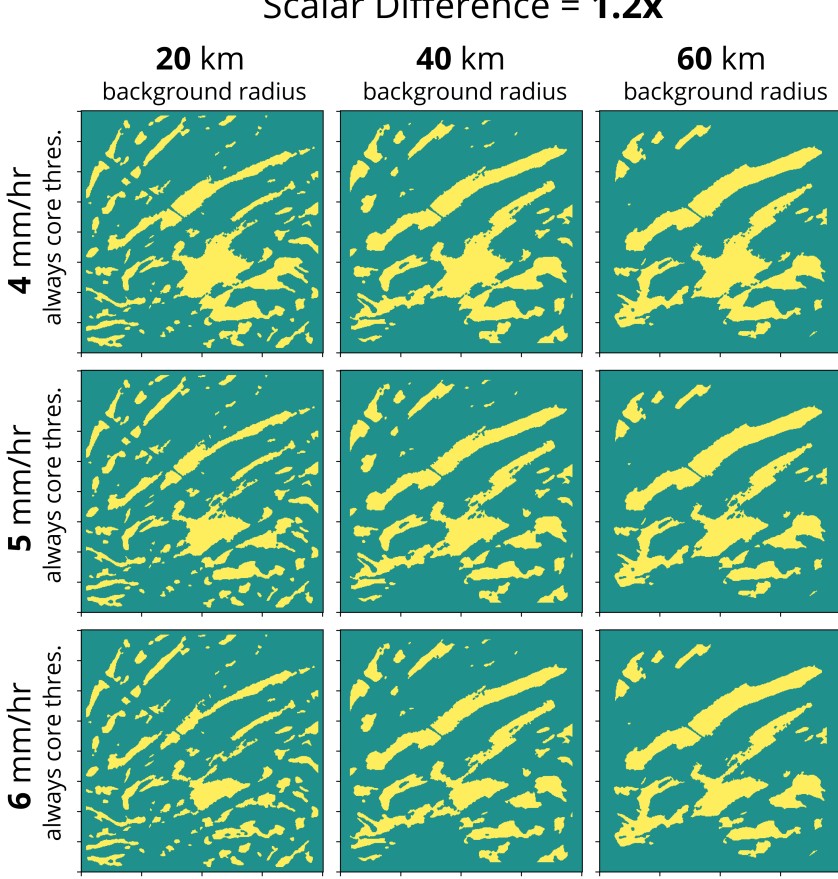

**Figure A2.** Example of how changing the tuning parameters impacts the feature detection output from the scalar scheme. Data shown is for a 401 km x 401 km grid with 2 km grid spacing from 7 February 2021 16:14:13 UTC from the Long Island, NY NEXRAD radar. The corresponding input reflectivity field is shown in Fig. A1a and the snow rate field is shown in Fig. A1b. The scalar multiplier of 1.2x is held constant in all 9 panels and the always core threshold and background radius are varied. Each column shows feature detection calculated with a 20 km (left), 40 km (center), and 60 km (right) background radius. Each row shows feature detection field calculated with an always core threshold of 4 mm hr$^{-1}$ (top), 5 mm hr$^{-1}$ (middle), and 6 mm hr$^{-1}$ (bottom).

## Scalar Difference = **1.5x**

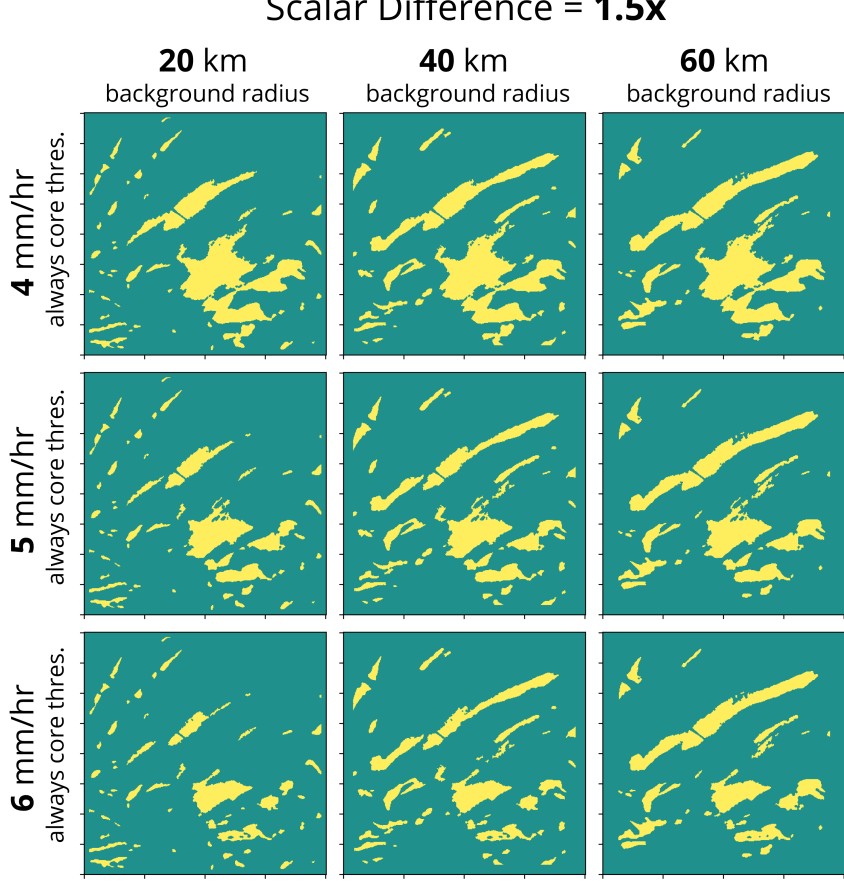

**Figure A3.** Example of how changing the tuning parameters impacts the feature detection output from the scalar scheme. Data shown is for a 401 km x 401 km grid with 2 km grid spacing from 7 February 2021 16:14:13 UTC from the Long Island, NY NEXRAD radar. The corresponding input reflectivity field is shown in Fig. A1a and the snow rate field is shown in Fig. A1b. The scalar multiplier of 1.5x is held constant in all 9 panels and the always core threshold and background radius are varied. Each column shows feature detection calculated with a 20 km (left), 40 km (center), and 60 km (right) background radius. Each row shows feature detection field calculated with an always core threshold of 4 mm hr$^{-1}$ (top), 5 mm hr$^{-1}$ (middle), and 6 mm hr$^{-1}$ (bottom).

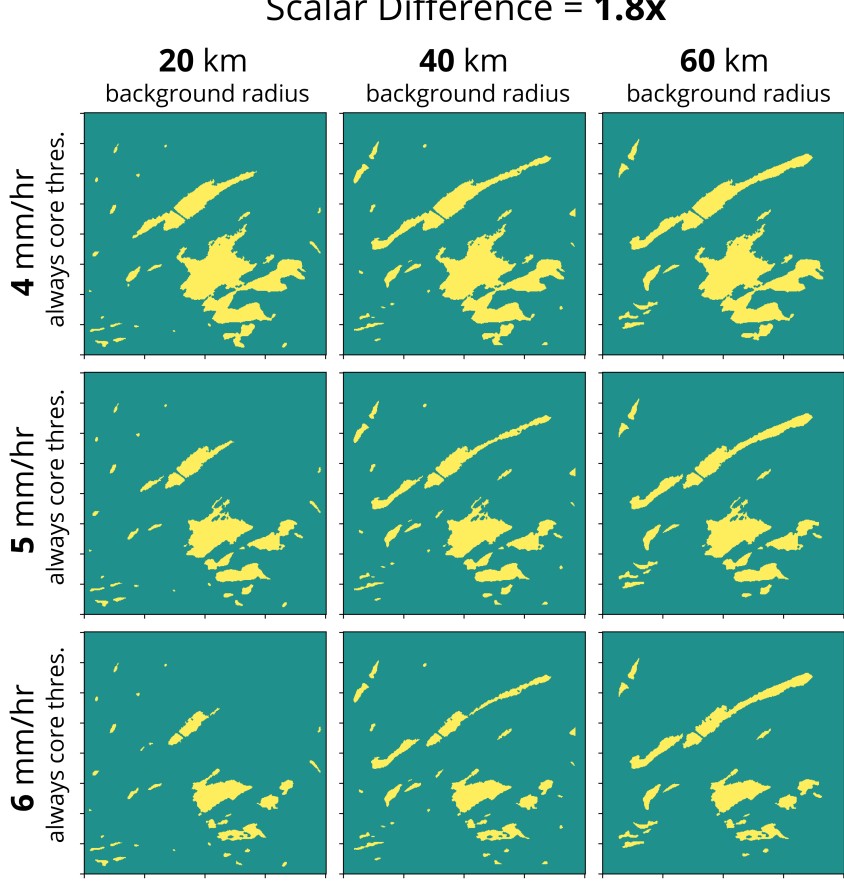

**Figure A4.** Example of how changing the tuning parameters impacts the feature detection output from the scalar scheme. Data shown is for a 401 km x 401 km grid with 2 km grid spacing from 7 February 2021 16:14:13 UTC from the Long Island, NY NEXRAD radar. The corresponding input reflectivity field is shown in Fig. A1a and the snow rate field is shown in Fig. A1b. The scalar multiplier of 1.8x is held constant in all 9 panels and the always core threshold and background radius are varied. Each column shows feature detection calculated with a 20 km (left), 40 km (center), and 60 km (right) background radius. Each row shows feature detection field calculated with an always core threshold of 4 mm hr$^{-1}$ (top), 5 mm hr$^{-1}$ (middle), and 6 mm hr$^{-1}$ (bottom).

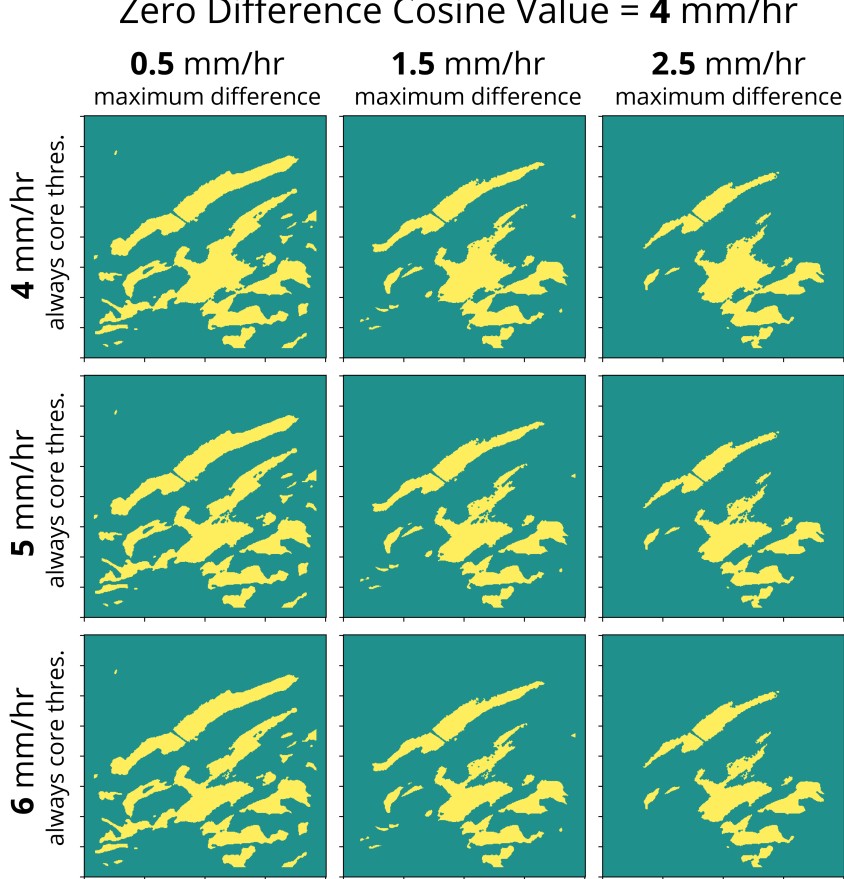

**Figure A5.** Example of how changing the tuning parameters impacts the feature detection output from the cosine scheme. Data shown is for a 401 km x 401 km grid with 2 km grid spacing from 7 February 2021 16:14:13 UTC from the Long Island, NY NEXRAD radar. The corresponding input reflectivity field is shown in Fig. A1a and the snow rate field is shown in Fig. A1b. The zero difference cosine value of 4 mm hr$^{-1}$ and background radius of 40 km are held constant in all 9 panels and the always core threshold and maximum difference are varied. Each column shows feature detection calculated with a maximum difference of 0.5 mm hr$^{-1}$ (left), 1.5 mm hr$^{-1}$ (center), and 2.5 mm hr$^{-1}$ (right). Each row shows feature detection field calculated with an always core threshold of 4 mm hr$^{-1}$ (top), 5 mm hr$^{-1}$ (middle), and 6 mm hr$^{-1}$ (bottom).

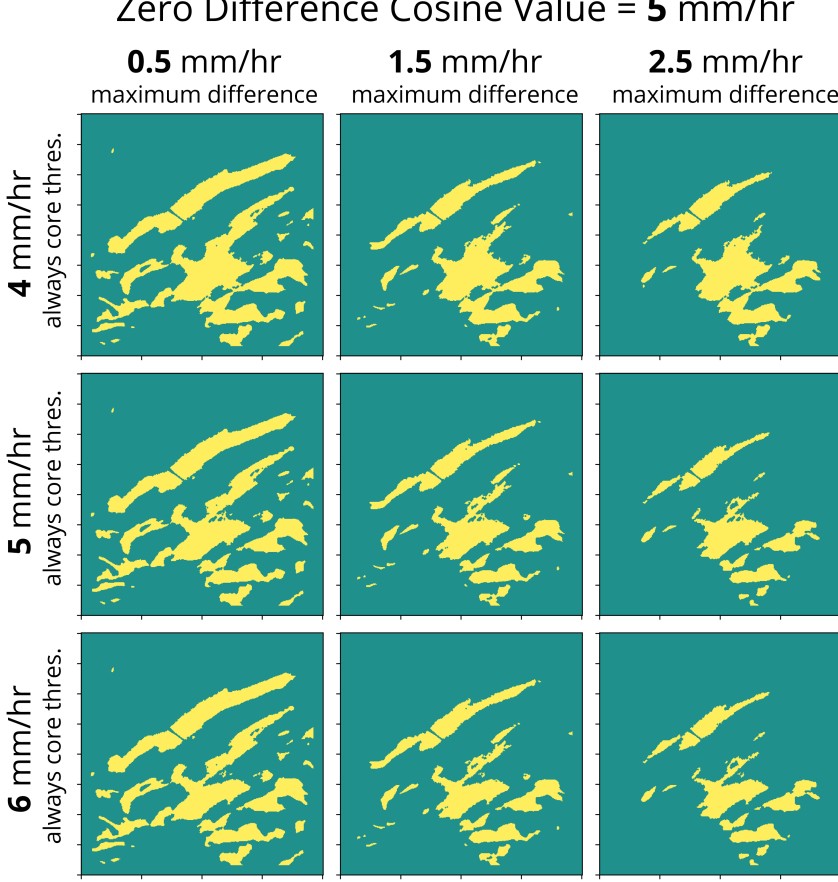

**Figure A6.** Example of how changing the tuning parameters impacts the feature detection output from the cosine scheme. Data shown is for a 401 km x 401 km grid with 2 km grid spacing from 7 February 2021 16:14:13 UTC from the Long Island, NY NEXRAD radar. The corresponding input reflectivity field is shown in Fig. A1a and the snow rate field is shown in Fig. A1b. The zero difference cosine value of 5 mm hr$^{-1}$ and background radius of 40 km are held constant in all 9 panels and the always core threshold and maximum difference are varied. Each column shows feature detection calculated with a maximum difference of 0.5 mm hr$^{-1}$ (left), 1.5 mm hr$^{-1}$ (center), and 2.5 mm hr$^{-1}$ (right). Each row shows feature detection field calculated with an always core threshold of 4 mm hr$^{-1}$ (top), 5 mm hr$^{-1}$ (middle), and 6 mm hr$^{-1}$ (bottom).

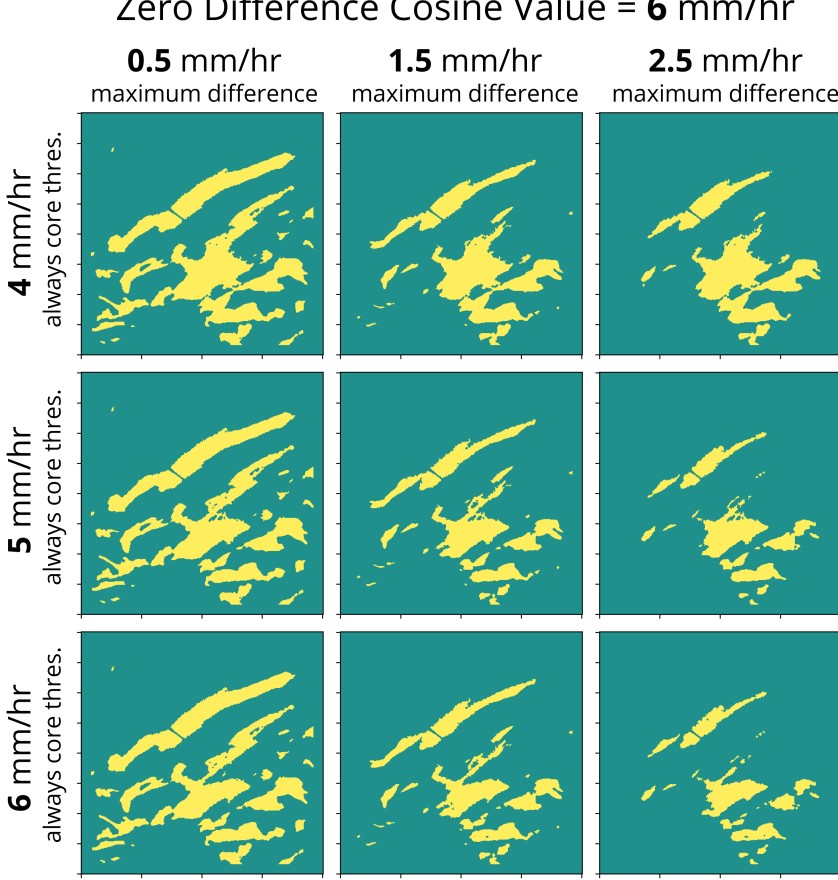

**Figure A7.** Example of how changing the tuning parameters impacts the feature detection output from the cosine scheme. Data shown is for a 401 km x 401 km grid with 2 km grid spacing from 7 February 2021 16:14:13 UTC from the Long Island, NY NEXRAD radar. The corresponding input reflectivity field is shown in Fig. A1a and the snow rate field is shown in Fig. A1b. The zero difference cosine value of 6 mm hr$^{-1}$ and background radius of 40 km are held constant in all 9 panels and the always core threshold and maximum difference are varied. Each column shows feature detection calculated with a maximum difference of 0.5 mm hr$^{-1}$ (left), 1.5 mm hr$^{-1}$ (center), and 2.5 mm hr$^{-1}$ (right). Each row shows feature detection field calculated with an always core threshold of 4 mm hr$^{-1}$ (top), 5 mm hr$^{-1}$ (middle), and 6 mm hr$^{-1}$ (bottom).

**Table 1.** Parameters used to detect locally enhanced echo features in winter storms. All input parameters to the algorithm as run for this paper are provided including those that are in effect turned off.

| Parameter | | Value |
|---|---|---|
| | Always core threshold | 5 mm hr$^{-1}$ |
| | Min. core size | 10 km$^2$ |
| Smoothing parameters | Background radius | 40 km |
| | Min. fraction for footprint | 0.75 |
| Cosine scheme parameters | Max. difference (i.e. $a$ in Eqn. 1) | 1.5 mm hr$^{-1}$ |
| | Zero difference cosine value (i.e. $b$ in Eqn. 1) | 5 mm hr$^{-1}$ |
| Scalar scheme parameters | Scalar difference (i.e. $c$ in Eqn. 2) | 1.5 |
| Core radii parameters (turned off) | Max. core radius | 2 km |
| | Value for max. core radius | 10 mm hr$^{-1}$ |
| Background echo classification parameters (turned off) | Min. value used | 0 mm hr$^{-1}$ |
| | Weak echo threshold | 0 mm hr$^{-1}$ |