# Peer review of "Dual adaptive differential threshold method for automated detection of faint and strong echo features in radar observations of winter storms"

_EGUsphere, 2023_

## Author Response (AR1)

**Reviewer #1**

**Summary:** This study presents a new method for automatically detecting snow bands in radar observations. The method is based on two adaptive thresholds that are dependent on the average background snow rate. One threshold increases linearly with increasing background, while the other decreases along a cosine function. The simplicity of this method is convincing. The code has been published and is publicly accessible, allowing a wide audience to access it. The manuscript is well-written and well-structured, with clear and informative figures. Multiple examples from various snow storms are presented, and the automatically identified features generally match what would be identified by eye. Overall, the manuscript is of excellent quality. I have only minor questions and comments, mostly regarding clarifications or suggestions for improvement. Therefore, I recommend accepting the manuscript for publication after addressing these minor comments.

**General comments:**

- I would appreciate a brief discussion of potential limitations. The term "artifacts" is mentioned multiple times (lines 105, 114, 167). Could this lead to problems in specific scenarios? Are there any situations where the method could potentially misidentify features? For example, could the method potentially "over-identify" features due to the algorithm's ability to detect very faint features? I understand that it is not always possible to objectively define snow bands based solely on reflectivity. However, this could be a topic for discussion.

  **We added a discussion of limitations in Section 2.4.**

- There are several parameters that can be defined by the user. How sensitive is the method's output to these parameters? Providing a figure (perhaps as a supplement) that shows how the method's results vary when the most important parameters are changed would be helpful for potential users.

**We added an appendix illustrating how changing some of the tuning parameters including background radius, threshold for always convective, and scalar difference changes the feature detection field.**

- Perhaps you could comment on the computational efficiency. I would expect this algorithm to be fast, even on large datasets. Is this the case?

  **The speed is largely dependent on the specific computer configuration (i.e. memory, CPU, etc.) On our computer it takes ~1 minute per reflectivity array (601 x 601) to run (most of the time is spent calculating the background average).**

  **Added this to text: "We made minimal consideration to the speed and efficiency of the algorithm. While the algorithm runs fairly fast (about 60 seconds per 601 x 601 size reflectivity input field on our servers), there is room for improvement in code efficiency and the potential for parallelization to be incorporated."**

**Specific comments:**

- Lines 17-20: This argument is not universally true for all convective identification algorithms. Some algorithms use simple reflectivity thresholds instead of gradients, as convective precipitation is generally characterized by high reflectivity (For example, TITAN: Dixon and Wiener, 1993). Therefore, the reflectivity gradient does not affect the outcome of these algorithms.

  Dixon, M. and Wiener, G.: TITAN: Thunderstorm identification, tracking, analysis, and nowcasting – A radar-based methodology, J. Atmos. Ocean. Tech., 10, 785–797, 1993

  **Thank you for your comment. We have rephrased the sentence to read: "Unlike convective cells in rain which usually have high reflectivities and a sharp reflectivity gradient between the cell itself and the background reflectivity, snow bands have weaker reflectivities, stand**

**out less from the background, and the edges of snow bands can gradually taper out, creating an irregular edge.”**

- Lines 44-48: Consider clarifying the advantages of your approach over other existing methods, such as Ganetis et al., 2019 or Radford et al., 2019, which are referenced earlier in the introduction. Based on my understanding of this section, other approaches that use adaptive methods to identify snow bands primarily focus on identifying strong snow bands rather than weak ones. If this is what you meant, consider rephrasing to make it more explicit. Currently, it is only indirectly indicated by the sentence in line 44. If that is not your intended meaning, please clarify what is lacking in their approaches that necessitates the use of your new algorithm.

    **We rephrased the sentence to read: “For some applications, detecting only strong snow bands is sufficient.”**

- Lines 49 and 89-93: It is not clear to me why an estimated snowfall rate is more suitable for detecting weak features than reflectivity. Is it because of the linear scale instead of the logarithmic scale? Why not just use reflectivity in units of mm6/m-3 instead of dBZ?

    **Yes, we used estimated snow rate because it is more linear in snow. We chose to use snow rate instead of linear Z since snow rate is more physically intuitive. We added the following: “We chose to use liquid equivalent snow rate rather than linear Z since it is more physically intuitive.”**

- Line 95: I think you mean reflectivity difference in dB, or reflectivity in dBZ.

    **Thank you for catching this, we rephrased to “We do not use the derived snow rates for quantitative estimates of precipitation, just as an alternative scaling factor to reflectivity in dBZ.”**

- Lines 104-105: I think a circular smoothing makes more sense than a rectangular one anyway, so I suggest removing the sentence arguing with "artifacts".

  **We rephrased the sentence so it reads: "We found that use of circular footprints produced better results than rectangular footprints."**

- Line 148: Is the "always core threshold" truly necessary? Based on my understanding of the scheme, any background average value above the "Zero Difference Cosine Value" (referred to as "b" in Equation 1) would be identified as a core regardless. Would it not be possible to adjust "b" to achieve the same or at least a very similar result?

  **Yes, it is true that one could use the "b" value to act like the "always core threshold". Having both variables provides the user with more flexibility. For example, if one wanted a flatter curve (i.e. a high zero difference cosine value"), an always core threshold would be necessary.**

  **Below is a figure which shows the shape of the curve when the always core threshold is greater than the zero difference cosine value. You can set the values such that the always core threshold isn't needed, but it is helpful to have and is necessary for the scalar difference scheme which increases with increasing background value.**

[Figure]

**We added the following text: "An absolute threshold like the always core threshold is helpful for identifying cores in regions where the background values are high. It is particularly useful for the scalar scheme and provides additional flexibility in turning. The zero difference cosine value can be used in place of the always core threshold in the cosine scheme."**

- Line 150: A threshold value of 4 mm/h is used in Figure 4. Consider adjusting the figure to match the text.

  **Figure 4 is designed to illustrate the tuning variables for the curves in general. These values are different for identification of convective precipitation in rain storms (see Table B1 in Yuter et al. 2005).
  In our implementation for snow bands we used an always core threshold equal to zero cosine value, this is described in Table 1.**

**Yuter et al. (2005):**
**https://journals.ametsoc.org/view/journals/apme/44/4/jam2206.1.xml**

- Lines 205-209: When is image muting applied? Is it part of the software package or is the user expected to remove mixed phase echoes independently? It is evident that it has been applied to Figures 9, 10, 11, and 12. What about the previous figures 3, 5, 7, and 8? This paragraph may be better placed elsewhere in the manuscript, such as at the end of section 2 or section 3.

  **The image muting is applied to the full examples after the algorithm is run and before plotting. We rephrased the sentence to read: "After we run the algorithm to detect features, we apply image muting (Tomkins et al., 2022) as a separate step independent of the feature detection algorithm to identify regions of mixed precipitation in the winter storms." Image muting was not applied to Figs. 3, 5, 7, 8 since these are zoomed in on the snow features to illustrate how the algorithm works. Image muting is part of the Py-ART package and an example of how to incorporate it into the feature detection plots is present in Part 2 of the example hosted on the Py-ART webpage.**

  **Imaging muting function:**
  **https://arm-doe.github.io/pyart/API/generated/pyart.util.image_mute_radar.html**
  **Example of incorporating to feature detection plots:**
  **https://arm-doe.github.io/pyart/examples/retrieve/plot_feature_detection.html#part-2-cool-season-feature-detection**

- Line 221: I do not understand how your algorithm minimizes flashing. This is not explained.

  **We added the following sentence: "The flashing of features occurs when a particular enhanced feature alternates between being detected and not being detected in sequential times. A key goal of the algorithm**

**development was to minimize flashing of individual features in consecutive times. Small speckles are more prone to flashing than larger area features which is why we filter out small objects. While we minimized flashing as best we could, there are still times when features are not consistent through time. Another aspect of flashing occurs where the edges of enhanced features can alternate between strong and faint.”**

- Line 255: What do you mean by “less precise reflectivity data”? Was the input radar reflectivity of other methods somehow not well measured? Or do you mean that using reflectivity instead of snow rate gives less precise results in general? I would disagree with the latter. If you are referring to previous studies using reflectivity on a logarithmic scale versus your snow rate on a linear scale, I would suggest that you refer to the scale rather than "reflectivity" here, since the word “reflectivity” can be used for both logarithmic (dBZ) and linear (mm6/m-3) units.

  **Here we mean that older methods often detected bands from Level-III reflectivity data which have precision of 5-dB.**

  **We rephrased the sentence to read: “Previous methods to automatically detect snow bands in radar observations either used inflexible thresholds and Level-III reflectivity data (5-dB precision) or used adaptive thresholds that were not able to detect objects that are not very distinct from the background.”**

- Figure 4: Please use the same y-axis for all three subplots.

  **Done.**

**Reviewer #2:**

The manuscript 'Dual adaptive differential threshold method for automated detection of faint and strong echo features in radar observations of winter storms'

describes a new method for feature identification of snow events in wintertime precipitation. It discriminates between faint and strong features. The manuscript is pretty well written, but some clarifications – mainly in the method description – are needed. The method seems sound, but some of the choices made need to be further justified. However, I believe that all of the issues listed below are easily addressed and I am recommending accepting the manuscript with minor revisions. The authors did a nice job overall. Specific comments are below.

- Line 6 in the abstract: It is not immediately clear to the reader what is meant by 'adaptive differential thresholds'. I would interpret adaptive thresholds as thresholds that change based on some objective criteria. Differential thresholds might be the same, or thresholds based on the difference between fields or times. A brief explanation of what you mean by 'adaptive differential thresholds' right up front in the abstract and again in the Introduction will ensure the reader is thinking of the right thing from the start.

  **Added the following text in the introduction to clarify: "Based on the difference between a pixel and the background average value, the algorithm determines if the pixel is part of an enhanced feature. The algorithm "adapts" the difference threshold to the mean background value.There are two ways a pixel could pass this test. One based on a criteria that requires a decreasing difference with increasing background value and one that requires an increasing difference with increasing background value"**

- Line 8 in the abstract: I am also not clear about what is meant by '…detects features within a snow rate field that is rescaled from reflectivity…' At this point we do not know how the snow rate is computed and to have it rescaled from reflectivity doesn't mean much. A brief, but clear description of what is meant here is needed.

  **We have rephrased the sentence to read: "The algorithm detects features within a snow rate field *rather than* reflectivity and**

**incorporates an under and over estimate of feature areas to account for uncertainties in the detection."**

- Lines 8 and 9: Same goes for '…an under and over estimate of feature areas to account…' It is not clear what is being estimated as over or under what? Again a brief clarification is needed here.

  **We have rephrased the sentence to read: "The algorithm detects features within a snow rate field rather than reflectivity and incorporates an under and over estimate *of feature areas* to account for uncertainties in the detection."**

- Line 19: The term 'feather out' seems colloquial and open to individual interpretation, which can muddy the intended meaning. I recommend using more precise language here.

  **We have rephrased the sentence to read: "Unlike convective cells in rain which usually have high reflectivities and a sharp reflectivity gradient between the cell itself and the background reflectivity, snow bands have weaker reflectivities, stand out less from the background, and the edges of snow bands can gradually *taper out, creating an irregular edge."***

- Line 36: I recommend describing what is meant by 'base reflectivity mosaics'. In my experience not all readers will know what this means. You could simply put in parentheses '(lowest elevation angle)' after base or something like that.

  **We rephrased the sentence to read: "Radford et al. (2019) used NEXRAD base reflectivity *(lowest elevation angle)* mosaics for three winter seasons and only considered objects that were 1.25 standard deviations above the mean reflectivity, as well as ≥ 250 km in length and with a minimum aspect ratio of 0.33 following the methods of Baxter et al. (2017).**

- Line 47: 'a automated' should be 'an automated'.

  **Thank you for catching this. We have updated the sentence to read: "In order to study these structures, we needed *an automated* snow band detection method that would detect a range of echo features from faint to strong."**

- Line 49: It states here that the method rescales the reflectivity. This is in contrast to the abstract that states the 'snow rate is rescaled from reflectivity'. This is very confusing and needs to be consistent and correct.

  **We rephrased the sentence to read: "This first step in the data processing *transforms* reflectivity to a value that is more linear in liquid equivalent snow rate."**

- Line 76: Python should be capitalized.

  **Thank you for catching this. We updated the sentence to read: "We contributed the software to the open-source *Python* package, Py-ART (Helmus and Collis, 2016), where it is available for general use."**

- Line 90: It is necessary to either show a reference or to show the proportionality listed here as dBZ ~ log10(mass3).

  **We added the reference to Matrosov et al., 2007, JAMC, which describes this proportionality https://journals.ametsoc.org/view/journals/apme/46/2/jam2508.1.xml**

- Lines 90 to 93: The statements that reflectivity is proportional to the mass^3 of ice (a single valued proportionality) and the snow rate can vary by a factor of 100 are logically incongruent. This passage needs to be rewritten and clarified.

**We rephrased the sentences to read: "A very rough first order approximation is that radar reflectivity dBZ $\propto \log_{10}(mass^3)$ for unrimed aggregates, where mass is the mass per unit volume of precipitation-sized particles (Matrosov et al., 2007). For rain, the radar reflectivity to mass relationship can be approximated by dBZ $\propto \log_{10}(mass^2)$. Multiple observational studies have shown that any one relationship between reflectivity and snow rate has high uncertainty since for given dBZ, the associated snow rate can vary by two orders of magnitude (Fujiyoshi et al., 1990; Rasmussen et al., 2003)."**

- Lines 49 – 50, lines 97 – 98 and Figure 3: Unless I missed something, the motivation to rescale reflectivity using the Z-S relation for wet snow is to produce a field with larger variations or contrast from the background to aid in feature detection. However, when I look at Figure 3, I see the reflectivity colorscale spans from 0 to over 40 dBZ, or 4 orders of magnitude in linear reflectivity units. The snow rate values range from about 0 to 10, or 1 order of magnitude. So even in its log units, reflectivity has a larger range of values than snow rate by a factor of 4 and if one considers linear units for reflectivity the range is 1000 times more than for snow rate. This is counter to the stated motivation for the Z-S rescaling and must be addressed in the manuscript. Relying on the figures is inadequate since the choice of colormap and plotting limits can have a large impact.

    **The motivation for rescaling reflectivity to snow rate was not to have larger variations but to represent the field in a more intuitive way and relate it to something physical. We added the following sentence: "We chose to use liquid equivalent snow rate rather than linear Z since it is more physically intuitive."**

- Lines 117 – 119: I don't follow the sentence that begins with 'Smaller values will mean…'. First, do you mean 'fewer features' rather than 'less features' will be detected with smaller values? Second, what is meant by 'larger values will include all regions of the echo'? It seems to me that with a minimum fraction

of 1.0 that more echoes would be eliminated because that criterion is not met. That is not consistent with what is stated here, unless I am misunderstanding. This sentence should be rewritten for clarity and accuracy.

**Thank you for bringing this to our attention, this material was not clear. Thank you for catching this. We rephrased to read "We use a minimum fraction of 0.75 (i.e. the footprint must contain at least 75% echo coverage to be used in the analysis). This is done to minimize small, spurious features on the edge of the echo. The effects of the 0.75 minimum fraction can be seen in Fig. 3 where there are differences between the more jagged echo outer edges in the snow rate field (panels d-f) compared to the smoother echo edges in the background field (panels g-i). Changing the minimum fraction acts to change how much echo must be present in the circular background footprint for a given pixel to be considered in the algorithm. A minimum fraction of zero would yield a background field with identical outer edges to the snow rate field."**

- First two paragraphs of section 2.2.3: I am not sure of the purpose of the quotation marks on various terms here. These don't seem to be quotes from someone or another document. I recommend removing the quotation marks from terms or wording that are not a direct quote.

**We removed the quotation marks.**

- Figure 4 and faint and strong features: The definitions of faint and strong features were given as "We define two varieties of locally enhanced features, those that are have smaller differences from the background, faint features, and those that have larger differences, strong features." There is no reference to the strength of the background field in this definition. However in the description of the cosine and scalar multiplier schemes and in Figure 4 it is clear that faint features as defined are only associated with weaker background values. This causes confusion and needs to be clarified in the

text.

**We rephrased the sentence to clarify: "We define two varieties of locally enhanced features, strong features that have larger differences from the background and faint features that have smaller differences from the background where the background field is weak."**

- Line 160 (and others): I am puzzled by the explanation that a detection threshold that increases with increasing background value 'helps to distinguish both the feathered edges of stronger features as well as features that differ only slightly from the background.' For any given situation the slope of the threshold is irrelevant, it only matters what the threshold is for that situation. If a threshold increased with increasing background, but started out with a large threshold it would not detect 'faint' features. The relevant information is that the threshold is small for small background features so it can distinguish faint features there. The increasing threshold changes the sensitivity to faint features with changing background values, but does not help distinguish faint features by itself. The impact of the increasing threshold with increasing background field is to make the method more sensitive to features that differ only slightly from the background in regions with small background values (weak precip) than in regions with large background values (stronger precip). This seems to be lost in the explanation.

**We rephrased the text to read "The scalar multiplier scheme uses a difference threshold that increases linearly as the background value increases (Fig. 4b). This allows the scalar multiplier to pick up subtle features that are not very distinct from the background when the background values are small (i.e. in regions of weak precipitation). For example, for a background value of 1 mm hr$^{-1}$, the cosine scheme threshold is 1.4 mm hr$^{-1}$ while the scalar scheme threshold is 0.5 mm hr$^{-1}$."**

- Section 2.2.3: It must be explained why one adaptive differential threshold scheme is linear and the other is not. The shapes of these curves must have a purpose and it needs to be explained.

  **We rephrased the text to read: "The cosine relationship has a decreasing threshold with increasing background value (Fig. 4a). The cosine scheme uses simple and intuitive parameters to define a smooth, curved relationship between the background value and the difference threshold. The scalar multiplier scheme uses a difference threshold that increases linearly as the background value increases (Fig. 4b). This allows the scalar multiplier to pick up subtle features that are not very distinct from the background when the background values are small (i.e. in regions of weak precipitation). For example, for a background value of 1 mm hr$^{-1}$, the cosine scheme threshold is 1.4 mm hr$^{-1}$ while the scalar scheme threshold is 0.5 mm hr$^{-1}$. After extensive testing on many idealized and real examples from winter storms, we found that a combination of both types of adaptive thresholds was needed in order to detect the full range of reflectivity features from faint to strong. The cosine scheme only identifies objects that are very distinct from the background, while the scalar multiplier scheme identifies objects that are both very distinct and not very distinct. We chose the particular equations described here as they were both intuitive and easy to tune."**

- Line 172: Why is the example shown in Figure 7 and described here only for the cosine scheme? I thought you used a combination of the cosine and the scalar scheme thresholds. If there is a good reason for this, then state it. Otherwise use an example that is consistent with what has previously been stated as the method being developed and used.

  **We included the scalar scheme examples in Fig. 7.**

- Line 192: I don't understand what is meant by 'For the underestimate, echo where the original reflectivity field ≤ 2 dB gets removed'. The term 'dB' is unitless and denotes only a change in some quantity, e.g. reflectivity. For example a reflectivity of 2 dBZ is 2 dB more than a reflectivity of 0 dBZ. So as written, this passage doesn't really make sense. This must clarified.

  **We rephrased the sentence to read: For the underestimate, echo where the original reflectivity field ≤ 2 *dBZ* gets removed, so the underestimate feature detection field will have less total echo area than the best and overestimate feature detection fields.**

- Line 192: if 'For the underestimate, echo where the original reflectivity field ≤ 2 dB gets removed' means that any echo with reflectivity less than 2 dBZ is removed, then this must be justified. If it means something else, then that must be justified. As written, it seems that data are being arbitrarily removed but the reader does not know why.

  **Because we are decreasing the entire field by 2 dB and we do not consider values ≤ 0 mm/hr, regions where the reflectivity is ≤ 2 dBZ become ≤ 0 dBZ and are no longer considered in the analysis. We rephrased the sentence to read: "For the underestimate, *since we do not consider values ≤ 0 mm/hr (Table 1)*, echo where the original reflectivity field ≤ 2 dBZ gets removed, so the underestimate feature detection field will have less total echo area than the best and overestimate feature detection fields."**

- Paragraph on line 205: The source(s) of the temperature data to determine the existence of mixed precipitation is should be described here.

  **We are not using temperature data. We are plotting regions with likely mixed or melted precipitation in a gray scale, using the radar correlation coefficient field following the methods described in Tomkins et al. 2022.**

**We added the following to the text: "After we run the algorithm to detect features, we apply image muting (Tomkins et al., 2022) as a separate step independent of the feature detection algorithm to identify regions of mixed precipitation in the winter storms. This step de-emphasizes portions of the echo that pass through the 0°C level by utilizing information from the correlation coefficient field. Regions where the reflectivity is ≥ 20 dBZ and the correlation coefficient are ≤ 0.97 are considered to be likely melting or mixed precipitation and are colored in a grayscale (Tomkins et al., 2022). The sharp temperature gradients in winter storms can yield mixed precipitation echo regions that resemble bands (e.g. Picca et al. (2014) their Fig. 2 and Colle et al. (2023) their Fig. 7) and it is important to remove these mixed phase echoes before interpreting the detected features as snow. Full details of how the image muting is applied can be found in Tomkins et al. (2022)."**

- Paragraph on line 205: This seems to be in the wrong section as it describes an important part of the method to eliminate non-snow echoes and doesn't belong in the Examples section.

  **Material moved to Methods section as requested (section 2.3).**

- Line 244: Does the radar data confirm the existence of melting snow? This is one of the strongest dual-pol signatures and so is should be easy to confirm that the muted regions are in fact melting snow. It would make a stronger argument to show this in the manuscript.

  **Material is now in Methods section. We provide a detailed analysis and explanation of the image muting technique in Tomkins et al. (2022) so we added the following sentence: "Full details of how the image muting is applied and evaluated can be found in Tomkins et al. (2022)."**